# Shifts in attention drive context-dependent subspace encoding in anterior cingulate cortex in mice during decision making

Márton Albert Hajnal[1] ✉, Duy Tran[2,3], Zsombor Szabó [1], Andrea Albert[1], Karen Safaryan [2], Michael Einstein[2], Mauricio Vallejo Martelo [2], Pierre-Olivier Polack [4], Peyman Golshani [2,5,6,7] ✉ & Gergő Orbán [1,7] ✉

Attention supports decision making by selecting the features that are relevant for decisions. Selective enhancement of the relevant features and inhibition of distractors has been proposed as potential neural mechanisms driving this selection process. Yet, how attention operates when relevance cannot be directly determined, and the attention signal needs to be internally constructed is less understood. Here we recorded from populations of neurons in the anterior cingulate cortex (ACC) of mice in an attention-shifting task where relevance of stimulus modalities changed across blocks of trials. In contrast with V1 recordings, decoding of the irrelevant modality gradually declined in ACC after an initial transient. Our analytical proof and a recurrent neural network model of the task revealed mutually inhibiting connections that produced context-gated suppression as observed in mice. Using this RNN model we predicted a correlation between contextual modulation of individual neurons and their stimulus drive, which we confirmed in ACC but not in V1.

With a limited capacity to process sensory stimuli in an information-rich world, how the brain distinguishes relevant from irrelevant stimuli is one of the core problems in neuroscience. Out of a torrent of sensory features the nervous system is required to select the limited few that predict the outcome of actions. Animals learn task contingencies by learning which stimuli predict positive outcomes. Beyond this basic form of learning, when the relevance of specific stimuli change, animals can learn different contingencies by associating specific cues to task contingencies. When such cues are available, learning simply occurs in the space of stimuli and outcomes, augmented with cues, such that cues determine the set of features to attend. One proposed mechanism of attention is the enhancement of the relevant and suppression of the irrelevant feature[1]. However, a fundamentally different

computational problem arises when there are no cues to unambiguously establish the set of features attended or when relevance of cues is volatile[2]. In these conditions, it is unclear how conflicting contingencies between stimulus and outcome can be resolved by attention.

To address this question, we adopted a set-shifting task[3–5] in which animals were presented with both visual and auditory stimuli, of which only one was relevant for obtaining a water reward, while the other acted as a distractor. Whether visual or auditory stimuli were relevant for obtaining reward was stable during a block of trials and was switched at block boundaries. Importantly, in this set-shifting task no external cues were provided to signal the relevance of the presented stimuli. Consequently, the task required the animals to cope with the

[1]Department of Computational Sciences, HUN-REN Wigner Research Centre for Physics, Budapest, Hungary. [2]Department of Neurology, David Geffen School of Medicine, University of California, Los Angeles, Los Angeles, CA, USA. [3]Albert Einstein College of Medicine, New York, NY, USA. [4]Center for Molecular and Behavioral Neuroscience, Rutgers University, Newark, NJ, USA. [5]Integrative Center for Learning and Memory, Brain Research Institute, University of California, Los Angeles, Los Angeles, CA, USA. [6]West Los Angeles VA Medical Center, Los Angeles, CA, USA. [7]These authors jointly supervised this work: Peyman Golshani, Gergő Orbán. ✉e-mail: hajnal.marton@wigner.hun-ren.hu; pgolshani@mednet.ucla.edu; orban.gergo@wigner.hun-ren.hu

changing relevance of stimulus modalities. Thus, in the absence of direct cues, animals inferred the contexts in order to resolve apparent conflicts by constructing an internal representation from stimulus, decision, and reward contingencies. This paradigm established a setting in which we could investigate the computations necessary under context uncertainty.

The anterior cingulate cortex (ACC) has long been implicated in monitoring conflicts[6], a computation critical for identifying the need to update stimulus-outcome contingencies. More recently, a range of studies have suggested that ACC plays a more active role in updating stimulus-outcome contingencies: (1) modulating its activity after attentional errors[7]; (2) modifying behavior after negative experiences[8]; and (3) improving value estimates for planning goal-oriented behavior[9–12]. Therefore, during the set-shifting task several studies have argued for a division of labor between PFC and ACC: while prefrontal cortex (PFC) collects action and outcome history[13], ACC is modulated by past information when conflicts induce errors[11,13,14]. Without a cue indicating changes in stimulus-outcome contingencies, even after learning the paradigm, the animal must actively monitor conflicts and establish an internal cue that identifies currently valid contingencies. Notably, active conflict monitoring is also critical before successful learning of the paradigm and therefore ACC is expected to have an important role during learning as well. Taken together, we hypothesize that by tracking conflicting situations, ACC represents internal cues that define the relevance of sensory features. Then, relying on these internal cues, ACC implements a mechanism that prioritizes information relevant for the outcome of actions by suppressing the irrelevant input, a mechanism we refer to as context-gated suppression.

To understand how attention mechanisms select the relevant features in a given context, we examined neuron population activity in ACC during the performance of the set-shifting task. We found that neural correlates of stimulus and context are represented in low-dimensional manifolds, such that linear subspaces can be identified with both stimulus and context, as previously observed[4,15,16]. We found that in this subspace the irrelevant stimulus modality was systematically suppressed during sensory stimulus presentation. In contrast, in recordings from V1, no such suppression could be identified. We identified context-gated suppression of the irrelevant stimulus as a mechanism that is well aligned with the representation found in ACC. Deriving analytical proof and modeling the set-shifting task with a Recurrent Neural Network (RNN), we reproduced key properties of our ACC recordings. Context-gated suppression was demonstrated in the RNN. This signature was in turn identified in neural recordings as well, underscoring the importance of context-gated suppression as a mechanism for establishing stimulus-outcome contingencies. Our work thus provides a mechanistic account of how stimulus relevance can determine our choices under conflicting evidence.

## Results

To dissect the neural mechanisms of attentional set-shifting we trained mice to perform an audio-visual cross-modal context-dependent decision-making task[4]. In this task, we simultaneously and randomly displayed one of two visual stimuli (45° or 135° drifting gratings) and one of two auditory stimuli (5 and 10 kHz tones). During each block of trials mice had to make "go"/"no-go" decisions (licking for water reward) based on one sensory modality while ignoring the other modality (Fig. 1a). Therefore, only one of the stimulus modalities was relevant for making a correct decision, while the other modality was irrelevant. During each trial, visual and auditory stimuli were presented for 3 s with the water reward available during the third second (Fig. 1b). Intertrial interval was 3 s but was extended to 9 s as a timeout for incorrect responses. Each block (50–150 trials) was preceded by a priming block (30 trials) where animals made perceptual decisions based only on a single modality presented. They then needed to continue to make sensory decisions based on that

modality during the subsequent cross-modal decision-making block of trials. Our paradigm design avoided explicit per-trial contextual cues that could identify the relevant stimulus modality. Therefore, mice had to learn stimuli, action, and reward contingencies together as a set to be able to infer context. Trials could be divided into congruent and incongruent trials. During congruent trials, both stimuli signaled the same decision. These trials were therefore not informative as to whether the animal was attending to the correct stimulus. During incongruent trials, one stimulus modality signaled the animal to lick and the other to refrain from licking, and as such incongruent trials implied a conflict between concurrently presented stimuli. Therefore in incongruent trials, the animal had to attend to the correct modality to make the correct decision.

Mice were first trained on the auditory modality, then on the visual modality, and finally on the compound two-modality context-switching task. Mice were trained on each stage until they were capable of performing in that stage at d'>1.7 (probability of chance behavior <0.1%). Animals performed 300–500 trials during training in the last stage. Note that during training longer sessions were performed than during the recording session. Only one training session was conducted per day with the aim of giving the animal all their daily water allotment during training. If animals did not receive their full allotment of water for the day during training, animals were given supplemental water an hour following training. Whether the animal started with the attend-visual or ignore-visual trial set was randomized. The training stopped when animals showed average standard deviation from chance performance d'>1.7 for both two-modality blocks. The detailed behavior of the animals from which V1 recordings were obtained were described previously[4]. We analyzed behavior by separating congruent and incongruent trials and calculating a 21 trial moving average for each of the "go" and "no go" trials separately (Fig. 1c and Supplementary Fig. 1a–d). We defined consistent task performance as parts of the session when the moving average fraction-correct of all four trial types was above 0.5. The number of trials in consistent periods varied from animal to animal, but consistent periods could be identified in all blocks (Fig. 1d, for n = 4 mice proportion of consistent trials 88/8, 18/35, 24/7, 82/45% for visual/auditory contexts, respectively). Animals tended to perform more reliably in consistent trials, in trials in which both stimulus modalities indicated the same behavioral outcome. We tested if the identified consistent periods resulted from selectively performing well on the congruent trials but relying on a context-unaware random licking strategy in incongruent trials. For this, we synthesized behavioral data with length and lick rate matching the empirical length and lick rates in incongruent trials, such that these empirical rates were specific to individual animals. We assessed the probability of consistent blocks under a distribution of synthetic choices for attend-audio and attend-visual blocks separately. We found that at all experimental blocks the hypothesis that the observed choices came from the context-unaware strategy could be rejected (Fig. 1e and Supplementary Fig. 1e–h). While all probabilities were under 0.05 significance level, smaller values indicated more extended periods of consistent task performance. Importantly, as we intended to assess the differences between the neural response patterns when the set-shifting task was properly executed and those when the animal failed to abide by the rules of the task, we could capitalize on periods when an animal was in exploratory periods.

Next, we assessed the behavioral patterns of individual animals in consistent periods. Consistent periods were characterized by a high fraction of correct responses in most animals (Fig. 1f). Responses of animals could be explained by context-aware models at higher likelihoods than context-unaware models (Fig. 1g, h for ACC and previously published for V1[4], "Methods"). In trials not identified with consistent periods, animals licked more randomly or with biased licking responses; these exploratory periods could feature evidence collection for context inference, reward maximization during uncertain internal task model, satiation, or fatigue (Fig. 1c, bottom).

To assess the neural correlates of decision-making under changing relevance of stimuli, 128-channel silicon probes were inserted for a single session of recording in either V1 and ACC, recordings were spike sorted with kilosort 2, and curated to exclude drifting units ("Methods").

We focused on delineating the neural dynamics underlying the computational steps preceding decision-making in the attentional set-shifting paradigm.

### Selective suppression of irrelevant information in ACC

In the attentional set-shifting task no immediate cues are available to indicate the set of relevant stimuli. Therefore tracking the behavioral relevance of different stimuli requires that outcomes of trials are constantly monitored. We analyzed the population activity in ACC to identify how behavioral relevance affects population responses. First we explored firing patterns of individual neurons during various task conditions (Fig. 2a, b). Cells exhibited varied responses between contexts and in responses to different stimulus conditions (from early stimulus time points to reward time point mean firing rate decreased 30–36% in 60–65 cells or increased 28–34% in 53–60 cells for each combination of visual and auditory "go" and "no-go" trials, out of $n = 122$ neurons from $n = 4$ ACC recorded animals). When assessing neurons with selectivity in firing to different visual or auditory stimuli, we found that these cells often responded with significantly different firing rates in the two contexts. When plotting firing rate trial averages relative to whether the stimulus was relevant or irrelevant in the

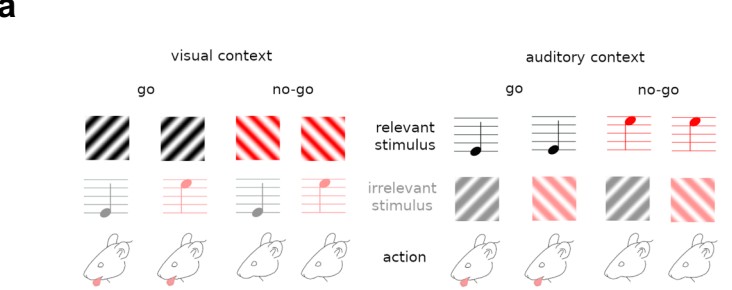

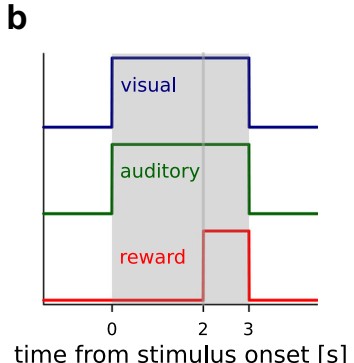

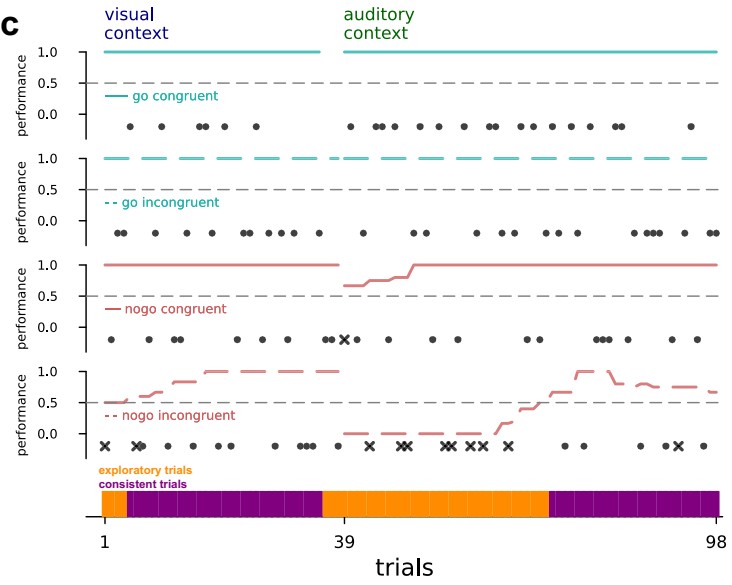

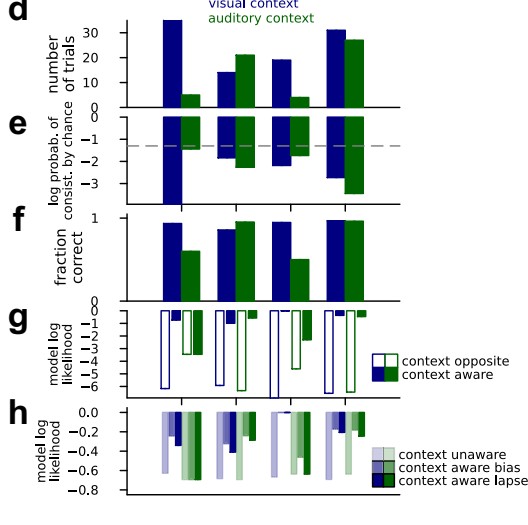

**Fig. 1 | Set-shifting paradigm. a** Stimulus and reward structure in the set-shifting paradigm. Visual (gratings) and auditory (pure tones) are presented concurrently to mice and animals. Only one of the modalities is relevant for obtaining water reward. One of the stimuli from the relevant modality (top row) was designated as a "go" signal, to which animals are expected to lick for water reward, while they were expected to withhold locking for "no-go" stimulus. **b** Time course of a trial, with simultaneous visual (blue) and auditory (green) stimulus presentation, and reward available from 2 s until stimulus end, represented as a pulse. **c** Behavior of an example animal in a visual to auditory context-switching session for different trial types (subpanels). Success and failure are indicated by black-filled disks and crosses, respectively. Lines show 21 trial equal weight moving averages. Trials were defined as "task-consistent" (bottom panel, purple) if the moving average performance of all four trial types were greater than chance, while other parts of the session were termed "exploratory". **d** Number of "consistent" trials for individual mice with ACC recordings ($n = 4$) for the visual (blue) and auditory (green) context. **e** Log probabilities of observing by chance the number of consistent trials in context blocks (colors), $10^6$ simulated sessions, where congruent trials were always successful, incongruent trials sampled from the Bernoulli distribution at $P$ equaling the empirical lick rate in incongruent trials of the mouse in that context. Significance level at 0.05 (gray dashed line). **f** Fraction of correct responses for all trials in the "consistent" periods. **g** Model log-likelihoods averaged over consistent incongruent trials in the visual (blue) and auditory (green) contexts for individual mice ($n = 4$) for a model targeting the opposite modality (empty bars), and the correct modality (filled bars). **h** Same as (**g**), but with a context agnostic model with mean choice lick bias (faint colors), and the context-aware model from the middle subpanel but augmented with either a bias or lapse parameter respectively (increasingly saturated colors).

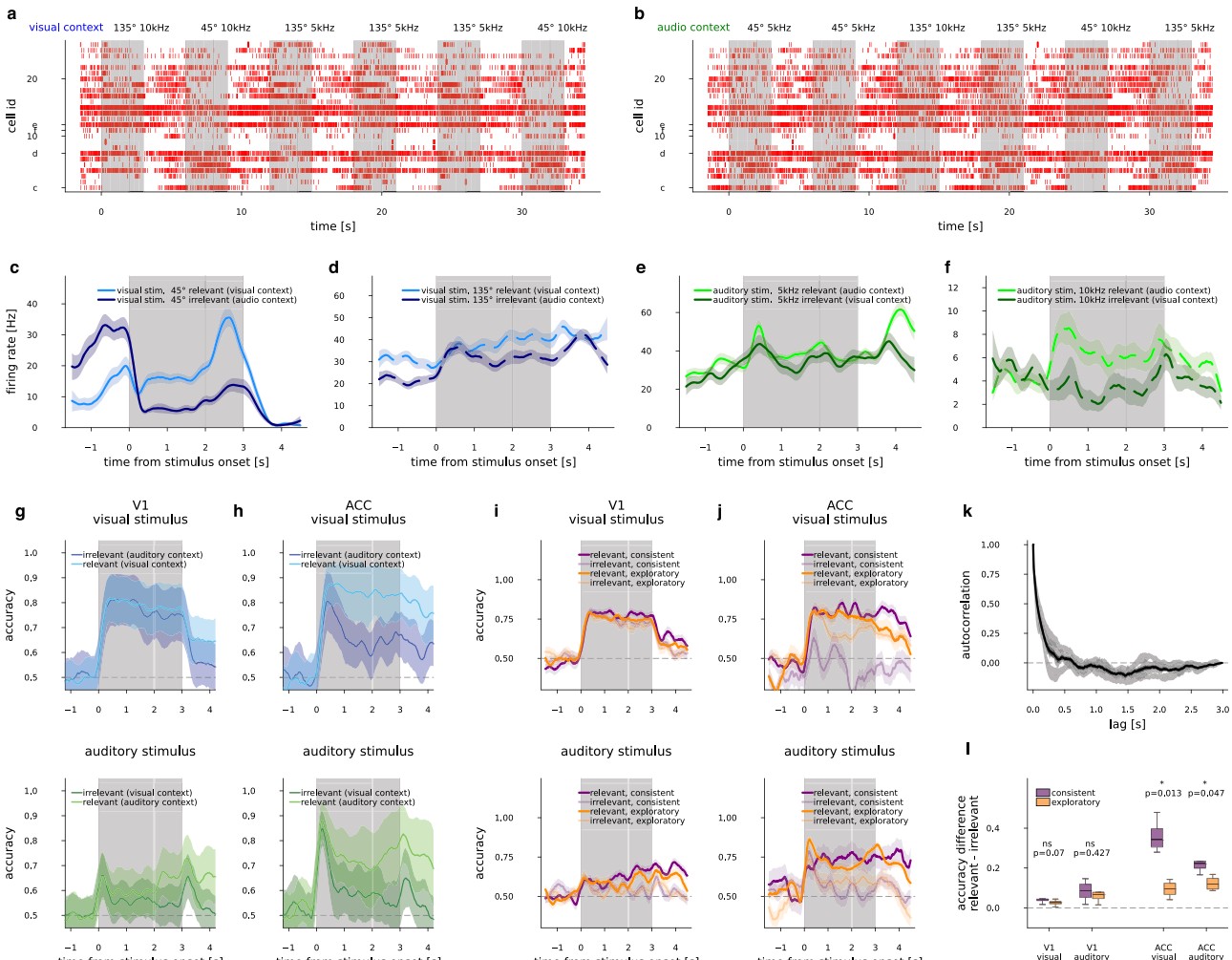

**Fig. 2 | Suppression of context-irrelevant stimuli. a** Spikes recorded from the ACC of an example animal during six trials (gray shaded areas), with various combinations of visual and auditory stimuli during the visual context. Cells are ordered by recording depth, cells labeled with letters c–f correspond to (**c**–**f**). **b** as (**a**), but from the auditory context. **c** Smoothed (300 ms window) mean firing rate of an example ACC cell ('c'), sensitive to visual "go" stimulus from visual "go" trials in the visually relevant (light blue) and irrelevant (dark blue) contexts, with shading indicating 1 s.e.m. **d** As in (**c**), but for an example neuron "d", sensitive to visual "no-go" stimulus from visual "no-go" trials. **e, f** As (**c, d**), but for neurons sensitive to auditory stimulus in audio-varied trials. **g** Visual (top) and auditory (bottom) stimulus decoding accuracies smoothed (300 ms window) in V1 along the course of trials in the relevant (light color) and irrelevant contexts (dark colors). Mean s.e.m. of $n = 8$ animals, with s.e.m. plus across animal mean of CV s.e.m. values. **h** Same as

(**g**), but decoding performed in ACC ($n = 4$ animals). **i** Visual (top) and auditory (bottom) stimulus decoding accuracies ($n = 8$ animals), as in (**g**), in the relevant (saturated colors) and irrelevant (faint colors) contexts for consistent (purple) and exploratory (orange) trials separately. **j** As (**i**), but in ACC ($n = 4$ animals). **k** Autocorrelation of decoder accuracies over time points, mean and s.e.m. of 8 conditions based on stimulus, relevance and consistence per mice (light gray lines and bands), and mean over mice and all conditions (black line and band). **l** Difference of decoding accuracies between relevant and irrelevant trials for V1 (left four boxplots, $n = 8$), and ACC (right four boxplots, $n = 4$). Distributions are calculated for 4 block-averaged time points in the 0.6–3 s time interval. Significance of the difference is displayed with two-sided $t$ test $P$ value estimates and confidence levels, boxplots feature the median, boxes at 25–75 percentiles, and whiskers extend to the minimum and maximum.

current context, it appeared that these cells fired at lower firing rates in the irrelevant condition in the 0.5 s period before water availability (Fig. 2c–f example neurons from one animal; 12, 10 cells for visual and 10, 15 cells for auditory "go", and "no-go" trials, out of $n = 122$ neurons from $n = 4$ ACC recorded animals, at 0.05 confidence level, see "Methods"). However, a similar number of neurons responded with opposite changes in firing rates between relevant and irrelevant conditions (24, 13 cells for visual and 10, 12 for auditory stimuli). While this modulation appears to be significant for around 20–35% of cells, as the direction of firing rate response changes were varied, single cell analysis provided limited investigative power. To investigate contextual modulation at a population level, we constructed time-dependent linear classifiers and decoded the visual and auditory stimuli identities from neural activity. Decoders were trained and cross-validated on

responses of ACC neurons in short time windows (50 ms), separate decoders were constructed for each subsequent time window ("Methods"). Thus, the decoder could track the quality of the representation of the two stimulus modalities as trials progressed. When decoding visual or auditory identity from ACC neuronal activity separately in the visual and auditory contexts, we found that both modalities behave similarly: irrespective of stimulus relevance, decoder performance was high following stimulus onset, but this was followed by a sharp decline in decoder performance when the modality was not relevant for making the decision. This was in remarkable contrast with decoding of visual stimulus identity in the primary visual cortex, where decoder performances did not display dependence on modality relevance (Fig. 2g, h and Supplementary Fig. 2a). We analyzed the stimulus-evoked responses of the population in more detail to gain

further insights into the observed decline in decodability of the stimuli. A decoder is defined by the hyperplane that optimally divides the population activities in response to different values of the decoded variable (identity of stimuli). Equivalently, the decoder can be characterized by the decision vector (DV), a direction that is orthogonal to the boundary, separating the identities of the decoded variable the best. Thus along the DV this view of the neural activity is reduced to one dimension, while separability is maximal between responses to different stimulus identities. Exploiting this property of DVs, we calculated trial-averaged population trajectories in "go" and "no-go" trials separately, then projected the absolute activity difference between "go" and "no-go" trial-averaged trajectories onto this DV. We found that the difference was often smaller in the irrelevant than in the relevant condition even early after stimulus onset for both modalities (4 context blocks out of 8 = 2 per 4 mice, Supplementary Fig. 2b). Thus, although irrelevant stimulus suppression can be active from the very onset of the stimulus, discriminability only gradually diminished throughout several seconds. We interpret the consistent drop in activity difference and decoder accuracy for irrelevant stimuli as context-dependent suppression of irrelevant information.

Since task context and thus stimulus relevance are only inferred from contingencies between stimulus, decision, and reward, the subjective relevance of a stimulus modality might change within a task context as well, should the animal lose track of context. If the observed suppression is a consequence of the animal's belief of the irrelevance of the given modality, we expect that in trials where suppression of the irrelevant modality is not sufficient to resolve a conflicting situation, mice would commit more errors. During a session, the behavior of mice was either characterized as "task-consistent", or "exploratory" in distinct blocks of trials (Fig. 1c, d). This gave us the opportunity to compare the difference between relevant and irrelevant stimulus decodability in the two behavioral states (Fig. 2i, j and Supplementary Fig. 2c, d). Less potent suppression was apparent in ACC during "exploratory" periods as opposed to "consistent" periods, in contrast with V1 where no dependence on behavioral state could be observed (Fig. 2l). Behavioral state dependence of the suppression strength between 0.6 and 3 s after stimulus onset was consistent across the animals in ACC in both modalities, with block averaging at 0.6 s windows where the window width corresponds to when the autocorrelation function relaxes to 0 (Fig. 2k, paired $t$ test, $n = 4$ time points * 4 animals = 16, $t = 5.3$, $P = 0.013$ and $t = 3.3$, $P = 0.0465$ for visual and auditory stimulus, respectively). No such dependence could be observed in V1 (paired $t$ test, $n = 4$ time points * 8 animals = 32, $t = 2.8$, $P = 0.07$ and $t = 0.9$, $P = 0.43$ for visual and auditory stimulus, respectively).

To investigate potential confounds, we introduced a number of controls. First, the observed difference in stimulus decodability might result from different levels of correlation between stimulus identity and decision in the two task contexts: when a particular stimulus modality is relevant, the stimulus identity predicts the decision, while when the modality is not relevant then predictive power of stimulus identity on decision is more limited. As a consequence, decision-related activity could carry information about stimulus identity in the relevant context. To control for this, we designed an analysis in which we conditioned trials for decision. Conditioning limited the number of available trials, thus certain combinations of parameters were not decodable (number of trials <10 per stimulus labels). As the number of no-lick trials was limited by a tendency to lick under uncertainty, we chose to perform the analysis on lick-only trials. Similarly, decoding from lick-only trials in consistent periods was severely limited as a consequence of fewer false alarms trials, thus we restricted our control analysis to exploratory trials. Stimulus decoding could be reliably performed from lick-only trials of exploratory periods in either the relevant, the irrelevant, or both conditions of all eight blocks of the four

animals. In these blocks stimulus decoding performance in lick-only trials closely followed decoding performance in unconstrained trials (all CV 2 s.e.m. bands overlapping, 10 out of 12 paired $t$ test $P > 0.1$, block-averaged between 0.6 and 3 s, Supplementary Fig. 4a–d), indicating that differences in movements have very small effects on stimulus decoding. Then, we explored suppression by contrasting relevant and irrelevant conditions. We found that in 3 out of 4 blocks where both the relevant and irrelevant decoders were available, suppression could be identified ($P < 0.03$ paired $t$ test, Supplementary Fig. 4b, c), in line with exploratory irrelevant trials showing smaller suppression than consistent trials.

Second, we investigated if the activity during the period in which suppression occurred had a distinct source from the stimulus-evoked activity recorded after stimulus onset. For this, we established the direction along which population activity changed when changing the stimulus (e.g., switching between 45 and 135° gratings). This one-dimensional subspace was used to project population activity during later stages of the trial, including the period of suppression. We hypothesized that if the source of activity during suppression was different, e.g., movement-induced activity, then the projected activity would be attenuated late during the trial both in the relevant and irrelevant conditions. We used population activity early after stimulus onset (0.25–0.75 s) and trained stimulus decoders on this early activity. As this time window corresponded to the time when stimulus-related activity reached ACC, the potential contribution of movement-related activity was severely limited. We used the DV of the stimulus decoder to project population activity onto the stimulus subspace and investigated if suppression can be identified in this stimulus-dominated subspace. The difference between population trajectories in response to two different stimuli was compared between relevant and irrelevant conditions (Supplementary Fig. 4e–h). We found that the difference between population trajectories was smaller in the irrelevant condition in 7 out of all 8 individual context blocks of the four analyzed mice.

Third, we performed single-neuron analysis to characterize the contributions of stimulus and choice to neuronal responses, and the suppression effect in particular. We linearly regressed the firing rates of individual neurons from either stimulus in its relevant context, choice or both at each time point. We performed this analysis in only consistent trials, as suppression was found to be significantly less potent in exploratory trials (Fig. 2l), although this in some cases severely limited the number of available trials for regression. We found that although the activity in most neurons in most time points cannot be predicted from simple binary variables, some stimuli were predictable, with a tendency for more probable predictability for relevant context (mean fraction of predictable neurons and time points in relevant context 0.10–0.25 and 0.00–0.08, while in irrelevant context 0.00–0.10 and 0.00–0.04 from all trials and from only consistent trials, respectively; proportions were largely symmetric across stimulus modality; in consistent trials 3 out of 8 context blocks from the four animals had at least one predictable neuron throughout most of the trial time course). We estimated the cross-validated variance explained in the population with the highest $R^2$ neuron at predictable time points, capturing the best representations available in the recorded population. The variance from choice and stimulus are not additive, as these variables are correlated. To control for choice and movement patterns potentially interfering with the estimation of suppression of the irrelevant modality and isolate variance differences on top of variance explained by choice, we compared estimated $R^2$s between the two-predictor models: choice + relevant stimulus vs. choice + irrelevant stimulus in each context. In all three predictable context blocks out of eight we found significant, 5–20% excess variance from relevant stimuli + choice compared to irrelevant stimuli + choice (all $P < 0.05$, paired one-tail $t$ test over 4 block-averaged time windows between 0.6–3 s, more details on Supplementary Fig. 4i–l).

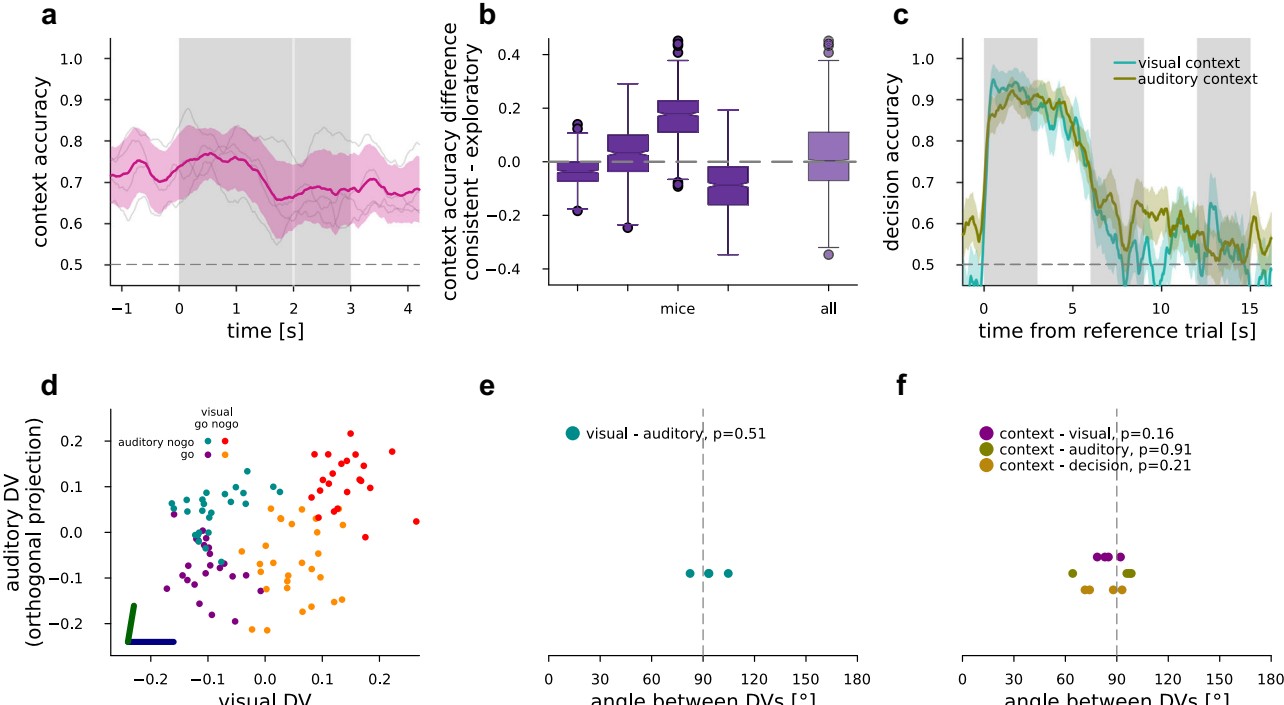

**Fig. 3 | Context inference, representation geometry. a** CV context accuracy in ACC for individual mice (gray faint lines) and mean and s.e.m. over mice (n = 4, magenta). **b** Context accuracy difference decoded from consistent over exploratory trials for each mouse individually (left) and the distribution for all time points for all mice (right, n = 4). Boxplots feature the median, notches at 95% C.I. for median, boxes at 25–75 percentiles, whiskers extend to 1.5 times the 25–75 percentile range. **c** Decoding decision from ACC neurons of an example mouse in the reference (leftmost gray vertical bar), and then two subsequent trials (all other gray vertical bars) within the sequence for each of visual (turquoise) and auditory (okker) contexts. Mean (lines) and s.e.m. (bands) over CV folds. **d** Population responses averaged over the first 0.5 s of stimulus presentation projected on the DV subspace in individual trials (dots), where colors correspond to visual and auditory stimulus pairing. Dark blue and dark green lines denote the DV directions of visual and auditory decoders, respectively. The horizontal axis is parallel to the visual DV, while the vertical axis represents the maximal projection of the auditory DV orthogonal to the visual DV. **e** Angle between the visual and auditory DVs in ACC for each animal (cyan dots, n = 4), mean of the first 0.5 s of stimulus presentation, P value of two-sided t test for mean over mice differing from 90°. **f** Same as (**e**), but between context and visual, auditory, or decision, respectively (purple, olive, light brown).

To summarize, context-dependent selection of the relevant stimulus is observable in mice, and successful selection of the relevant modality is correlated with the success of task execution.

## Geometry of population responses in ACC

In order to understand how the observed dynamical suppression is related to executing the task, we investigated the geometry of the representation of different task-relevant quantities in ACC. Since identical stimuli are presented to the animals in the contexts and correct execution of the conflict trials require opposing decisions in the two contexts, context (whether the animal should base its decision on visual or auditory stimuli) needs to be inferred from task contingencies and represented in the activities of the neurons. We constructed time-dependent linear decoders to identify task context in population responses (Fig. 3a). Context could be reliably decoded throughout the trial but reliability varied across animals (Fig. 3a). Since trials from different contexts are separated by a longer delay, the context signal might be contaminated by residual effects of electrode drift. To control for this potential confound, we performed an analysis using a specific cross-validation scheme: we trained context decoders in trials either at the edges of the entire session, or in the middle of the session, and tested reciprocally ("Methods"). We found no difference in decoder accuracy time courses between train-test directions (P > 0.1 in all mice, n = 10 in trial time course blocks, Supplementary Fig. 3a–d), signifying that the context signal identified by the context decoder has not emerged as a consequence of a continuous shift of uncontrolled variables. The strength of context representation in V1 was previously shown to be stronger in behaviorally consistent periods[4]. We investigated such a dependence in our ACC recordings as well. We did not find an improvement in context decoding in consistent trials vs. the exploratory trials (Fig. 3b).

To track the context successfully, the animal needs to keep track of the history of trial outcomes. ACC had previously been associated with representing the history of recent outcomes across trials when prediction error was high[13]. We asked if ACC populations could encode the choice the animal made in a particular trial across multiple trials. We used ACC population data to construct a time-dependent decoder for the choice the animal is making. To check the representation of a choice across subsequent trials, the time-dependent decoder was not constrained to the trial in which the animal made the choice but was extended into the intertrial interval and upcoming trials too. We found that ACC maintained information about these choices both during the intertrial interval and early in the upcoming trial too (Fig. 3c).

To understand how the neuron population concurrently represents the context variable along with stimuli of different modalities, we investigated the relationship of the representations by comparing the geometrical properties of the decoders. The two decision vectors (DVs, the best-separating directions in neural activity space) calculated for the auditory and visual modalities can be used to establish a two-dimensional subspace to contrast representations ("Methods"). Projecting neural response intensities (mean over the first 500 ms after stimulus onset) measured in individual trials into this two-dimensional subspace reveals that the auditory and visual stimulus-induced changes in population activity span close to independent subspaces,

highlighted by the close to orthogonal setting of the decision vectors (Fig. 3d). Such orthogonal representations ensure that different signals can be maintained without interference. This relationship between the visual and auditory decision vectors was consistent across animals (Fig. 3e, $n = 4$, one-sample $t$ test, mean compared to 90°, $t = 0.75$, $P = 0.51$). We found a similar orthogonal relationship between the context and the sensory or decision variables, which also held across animals (Fig. 3f, $n = 4$, $t$ test, mean compared to 90°, $t$ statistics were −1.87, −0.12, −1.59, $P$ values were 0.16, 0.91, 0.21 for context to visual, to auditory and to decision, respectively).

In summary, task variables are all represented in orthogonal subspaces.

## Set-shifting task in a recurrent neural network

Yet, the critical question remained as to how such orthogonal representations support set-shifting. To gain further understanding of the computations required to solve the set-shifting task, we developed a recurrent neural network (RNN) computational model that mimicked the stimuli-action-reward dynamics of mice performing the task (Fig. 1b). The input to the hidden layer consisted of stimuli in the current trial as well as a variable that tracked whether the previous trial was rewarded or punished (Fig. 4a). This information has previously been shown to be represented in the prefrontal cortex[13] and is present in our own experimental data outlined above. We optimized the network dynamics so that it was detailed enough to explore sequential computations and memory, but coarse enough that oscillations did not typically arise at the end of the training. 30 hidden neurons yielded robust capacity for memory and computation, with input and output variables encoded in neuron pairs (one-hot encoding). Input variables consisted of the following: (1) visual input; (2) auditory input; (3) reward or punishment for correct or incorrect choices, respectively. The RNN only had hidden weights and recurrent connectivity, but no gates. A time resolution of a nine-timestep-long stimulus presentation was chosen with three pre- and three poststimulus steps without timeout on errors. The network had to decide on the response during the stimulus presentation at timestep seven, and after the seventh timestep reward or punishment was delivered until the next trial's stimulus was presented. A single data point consisted of five subsequent trials from the same context in a sequence: four trials as a batch of trials during which the model had the opportunity to infer context, and a fifth final trial where the loss function was calculated from the last decision and fed to gradients for backpropagation. Training data was generated combinatorically for two choices by two modalities in 5-trial sequences, reaching 1024 unique sequences total with all possible trial combinations per context ("Methods").

The task design implies that maintenance of a context variable is necessary for successful task execution. Assuming that once the correct modality has been selected, the responses will all be correct, we expect that the proportion of trials where context is identified will be the same as the proportion of trials with correct decisions. Indeed, decoding context from the hidden units of RNN models, we found that the fraction of correct choices in incongruent trials were the same as the fraction of trials where the context was decoded correctly. Furthermore, after a typical performance drop in early training (epoch <200) due to strongly attuning to one of the modalities while performing poorly in the other modality, context representation and choice gradually improved in parallel, as performance became more and more context-symmetric (Fig. 4b). We observed a systematic difference in the rate of acquisition of congruent and incongruent trials: the RNN tended to perform congruent trials efficiently earlier than incongruent trials. This is consistent with the behavior of mice, which displayed asymmetrical performance in congruent and incongruent trials.

Based on normative considerations, congruent trials, where both modalities indicate the same action, the animal has no need to rely on

context information. Making a decision based on any of the two stimuli would lead to the same "go" or "no-go" choice. A non-trivial consequence of this is that no stimulus-action-reward contingency information could be used in congruent trials about the context, as the instructions from two modalities are indistinguishable. In contrast, an incongruent trial, indifferent to whether the decision of the animal is correct or incorrect, always gives information about the current context: The reward, or lack thereof can be compared with the stimulus and response. We can observe this phenomenon in the RNN models, where a large number of sequences are available with a fixed order of trial congruence (Fig. 4c). Note, that in the RNN data sequence setup, one sequence type of four congruent and one final incongruent trial cannot be solved, setting a theoretical limit to maximum performance on the last trial.

In the following, we investigate the representation and computations in RNNs trained on the set-shifting task to establish similarities with the properties of neuron populations recorded in ACC. Our electrophysiological recordings in mice showed that ACC neurons maintain the animals' choice beyond the termination of a trial (Fig. 3c). To assess if the RNN permits similar across-trial integration, we trained a linear decoder for choice. Corroborating experimental observations, we found the prolonged representation of the choice the model made in a particular trial across the intertrial interval and into the upcoming trials (Fig. 4d).

To contrast the representation of task variables in the RNN with that identified in the ACC of mice, we calculated decoder DVs for task variables. We found similar orthogonality to that of mice. Visual and auditory stimulus subspaces were orthogonal at the first stimulus point, as well as context DVs to stimuli. In addition, decision and context were also orthogonal at the time point of choice ($n = 88$, one-sample $t$ test, mean compared to 90°, $t$ statistics were −1.45, 1.17, 0.32, 0.11, $P$ values were 0.15, 0.24, 0.75, 0.91 for visual compared to auditory, and context compared to visual, to auditory and to decision, respectively, Fig. 4e, f). Thus, task variables were represented in analogous geometry in mice and our RNN model.

We investigated the dynamics of the trained RNNs to see how selection of the relevant stimulus modality was achieved. We projected activity in relevant and irrelevant contexts over the respective stimulus input weights, and similar to mice (Supplementary Fig. 3b), we observed suppressed activity of the irrelevant stimulus (Fig. 4g). This suppression was present at the first time point when stimulus was presented, a likely consequence of maintaining contextual information throughout several trials even between stimulus presentation periods (Fig. 4c), i.e., ready to engage when stimulus arrives. We then investigated how the abstract, context-sensitive output is computed based on the observed modulation of stimulus-related activity. We investigated the effective output of the hidden layer, by assessing the output projection weights of the hidden units to gauge their contribution to the decision of the RNN. As expected, the stronger the weight of a hidden unit to the output neuron was, the more its activity displayed context-dependent modulation of stimulus-related response: showing enhanced response for context-relevant stimulus, while negative response to context-irrelevant stimulus (Fig. 4h). Within-trial dynamics were characterized by a gradual buildup of the activity after stimulus onset towards the time point when the decision was made. RNNs also exhibited a strong correlation between decision performance and the modulation of the relevant and irrelevant context, expressed as the activity difference between the two contexts in response to the same modality in the cells with strong output weights ($n = 100$, $t = 7.37$, $P = 4.57 \times 10^{-12}$, Fig. 4i).

In summary, our RNN model reproduces several key properties of the neuron population recorded in ACC in the set-shifting task, such as the geometry and dynamics of the representations of the stimuli and the context. The model learned to infer the context from incongruent trials, and to suppress the irrelevant stimulus modality.

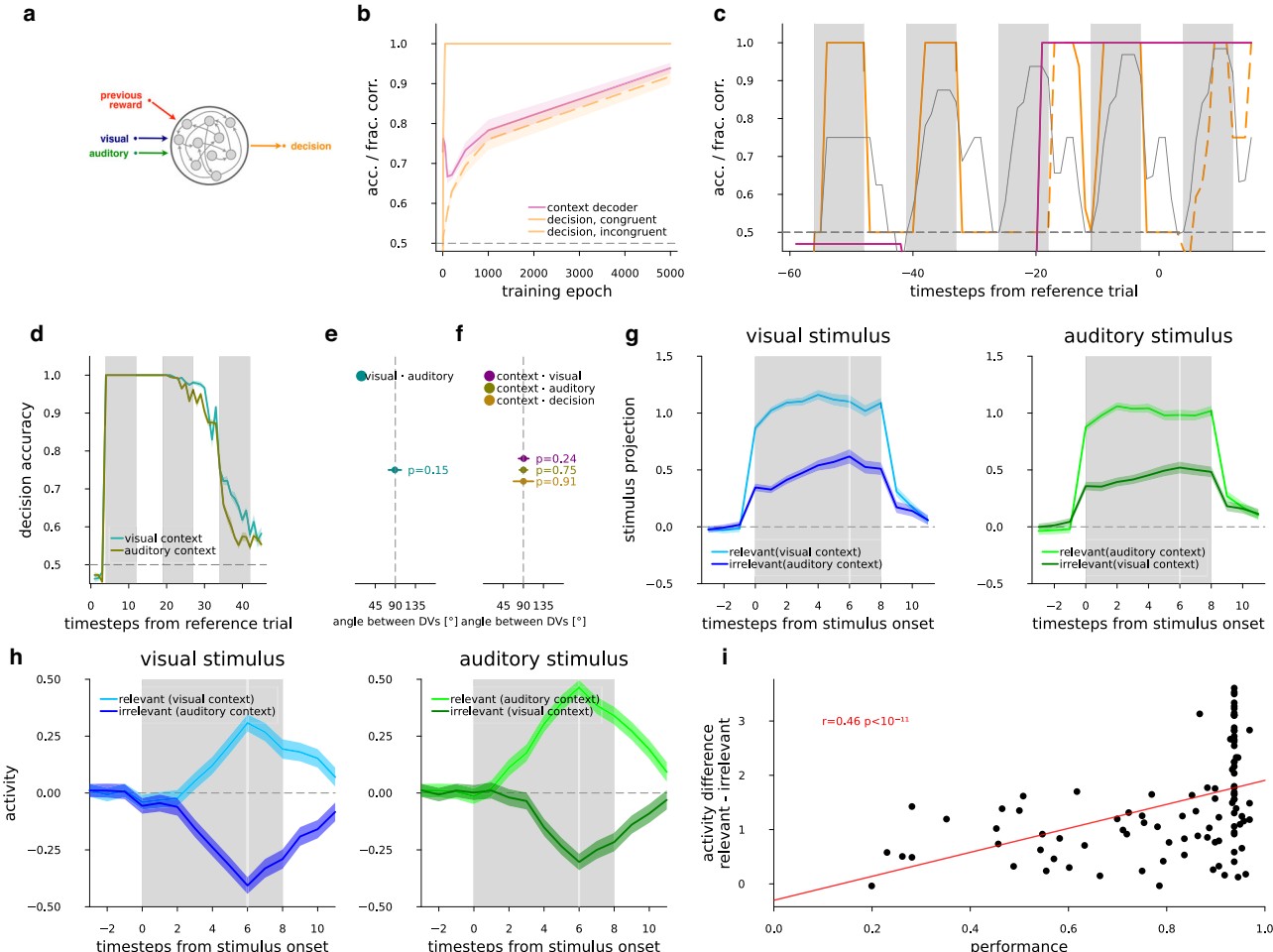

**Fig. 4 | Set-shifting task in RNNs. a** Schematics of the RNN model $\mathbf{h}_t = f(a_t, v_t, r_{t-1}, \mathbf{h}_t)$ and $d_t = f(\mathbf{h}_t)$ at the time resolution of trials (index t, see within-trial sequence equations in "Methods"), where the decision is a map to the decision space of "go" and "no-go", and is a function of the hidden state vector. The update to the state vector depends on the current visual and auditory stimuli, the previous hidden state, and the reward. Stimuli and reward were switched on and off at a fine time resolution with pulse envelopes as described in Fig. 1b. Note that reward during and after a trial depended on the preceding decision, which resulted in that prior to any decision made only the previous trial was presented (hence the label "previous reward"). **b** RNN performance (fraction-correct), and quality of the learned representation (context decoder accuracy) as the training of the RNN progresses (epochs). Average (lines) and s.e.m. (bands) of 20 RNNs for context accuracy decoded from hidden units (magenta), and fraction of correct responses (orange) in congruent (solid line) and incongruent (dashed line) trials over learning epochs. **c** Evolution of the decision signal and the task context when a trained example RNN is performing a sequence of five trials. Decision fraction-correct over all sequences (thin gray). Decision fraction-correct (orange) and context accuracy (magenta), for sequences with an example congruence pattern (c-c-i-c-i, where c congruent and i incongruent trials are indicated on the orange line by solid and dashed style, respectively, as in (**b**)). Vertical gray bars indicate stimulus presentations.

**d** Decoding decision from hidden units of an example RNN in the reference (leftmost gray vertical bar), and then two subsequent trials (all other gray vertical bars) within the sequence for each of visual (turquoise) and auditory (brown) contexts. **e** Mean (dot) and 2 standard deviations (not s.e.m., horizontal lines) of the angle between the visual and auditory DVs over the RNN models (cyan dot, $n = 88$ with performance >0.9), at the first time point of stimulus presentation, $P$ value of two-sided $t$ test for mean over models differing from 90°. **f** Same as (**e**), but between context and visual (purple), auditory (olive), or decision (light brown). **g** Activity from the two contexts projected onto the stimulus (visual: left, blue, auditory: right, green) input weights (equivalent to stimulus decoder DV), go and no-go trials separately, then averaged, mean (lines) and s.e.m. (bands) over models ($n = 88$ with performance >0.9). Stimuli are separately trial-averaged when they are in their relevant (light colors) and irrelevant (dark colors) context. **h** visual (left, blue) and auditory (right, green) stimuli when they are relevant in their respective context (light colors), or irrelevant in the opposite context (dark colors), responses of the $n = 2$ ("go" and "no-go") hidden neurons projecting to outputs with strongest weights, combined, mean and s.e.m. from $n = 88$ RNN models. **i** Regression between performance of a model on all trial types (horizontal axis) and difference between activities in relevant and irrelevant context for all stimuli combined from (**e**) for $n = 100$ RNN models.

## Context-gated attention in activity subspaces

Having reproduced some of the characteristic behavior of the recorded ACC neuron populations, we concluded that the RNN could be a promising tool to investigate how these observed representations may support set-shifting. For this, we first considered the attentional selection problem, then we identified a potential computational mechanism that can drive attentional selection, and finally we derived experimentally verifiable predictions.

Selection by attention can be regarded as a way to perform cognitive abstraction. The selection of the relevant quantity is not trivial

since (i) dimensional reduction needs to occur to map the space of all possible stimuli with various combinations of simultaneous, potentially conflicting instructions to a single-dimensional final action space, and (ii) this mapping depends on the context. To simplify the argument, initially we assume that the information about the context is already available in the animal and later we relax this assumption. We also assume that the task has already been learned, and connection weights do not change.

In the space of population activity, we describe the evoked differential activity between instances (i.e., "go" and "no-go") of the visual

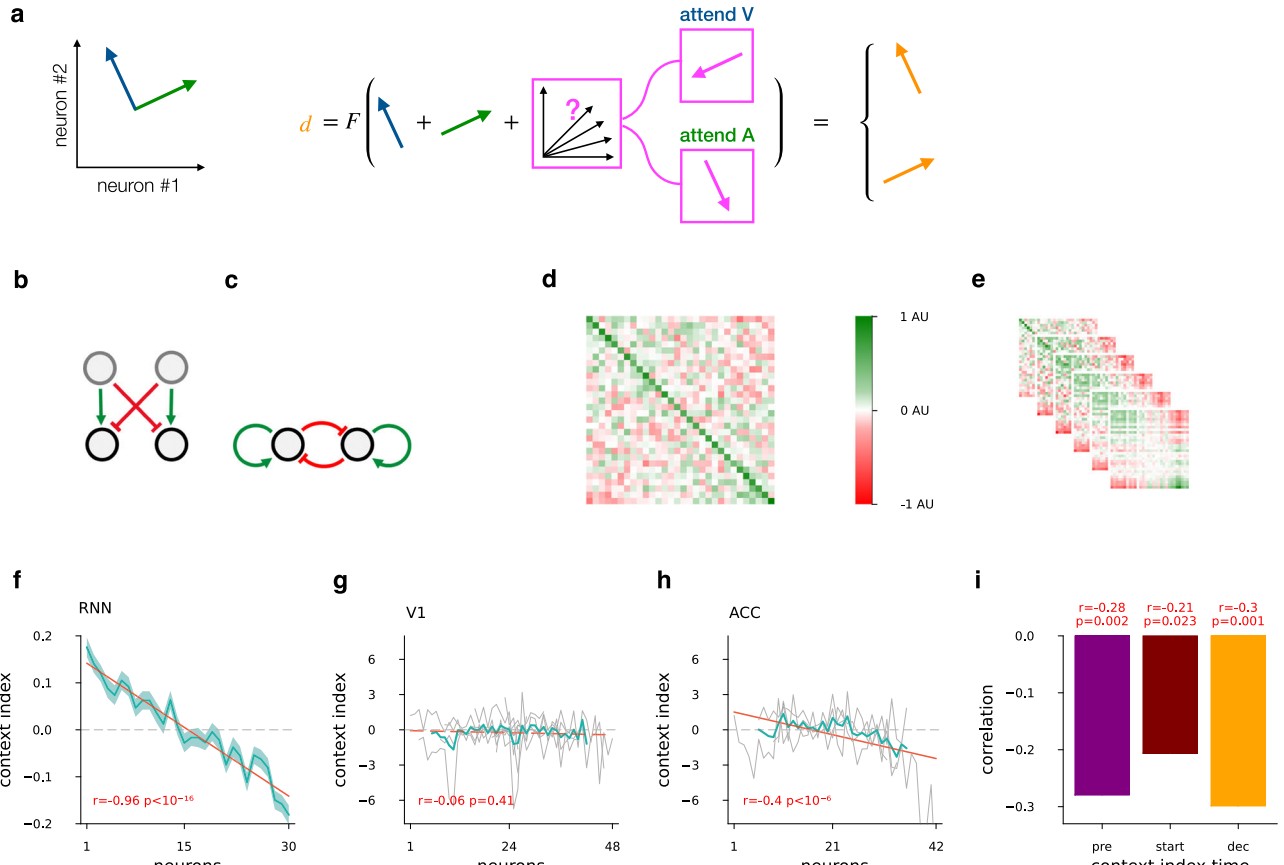

**Fig. 5 | Context-gated mutual feedback mechanism. a** Schematics of a context-selective inhibition mechanism when representations can be approximated by linear subspaces. The total stimulus space consists of unrestrained activities along the two subspaces (blue and green, left). In order to map the total stimulus space to the abstract decision space (orange, right) in a contextually correct way with fixed weights, the activity in the irrelevant subspace has to be inhibited (pink, middle). **b** Schematics of cross inhibition and selective amplification as a map from the context subspace (gray circles) into the total stimulus subspace (black circles): two input neurons innervate two neuron populations that represent activity in the two stimulus subspaces. **c** Schematics of the network that performs sequential processing for context-gated suppression. Context is represented together with the stimulus modalities in distinct subspaces (black circles) and mutual feedback controls the two subspaces. **d** Recurrent weight matrix of RNN models; neurons ordered by their modality index defined as input weight difference between visual and auditory stimuli. Colormap rescaled by the largest of absolute minimum or maximum (AU). Neurons ordered separately for "go" and "no go" signals, then averaged, mean of $n = 88$ models. **e** Assessment of sequential processing of information in the RNN through iterating the connection matrix. The 2nd to 6th power of the recurrent weight matrix on $d$ is shown, approximating its effect from stimulus onset until decision. Colormap rescaled, units as in (**d**). **f** Context sensitivity of neurons against their sensitivity to stimulus modalities in the RNN. Context index is defined as the difference of neural activity between the visual and auditory context; neurons ordered by the modality index, for RNN models, using input weight modality index as in (**d**, **e**), mean and s.e.m. over models, $n = 88$ that have performance >0.9 fraction-correct; Pearson-correlation ($r$ different from 0, two-sided $t$ test $P$ values). Values calculated and neurons ordered separately for "go" and "no go" signals, then averaged. **g**, **h** Context sensitivity of neurons against their sensitivity to stimulus modalities in recordings from V1 ($n = 8$ animals) and ACC ($n = 4$ animals) of mice, respectively. Recordings in different mice featured different numbers of neurons, with the neuron index set at the midpoint of the number of neurons for display purposes, skipping 6 start and end neuron order positions on each side. Context index was established based on the whole trial. Individual animals (gray), mean over mice (turquoise), linear fit to mean over mice (red). **i** Correlation between modality rank and context index, at three time intervals (colors, timing shown on top): before stimulus start ("pre", from −1.00 s to −0.25 s), after stimulus start ("start", from 0.0 s to 0.75 s) and at the required decision time ("dec", from 1.75 s to 2.50 s) in ACC.

or auditory stimuli by vectors **v** or **a**, respectively. Thus, the activity arising as a consequence of joint presentation of the two stimuli is simply **v** + **a**, which resides in a space that is a direct sum of the orthogonal subspaces spanned by the visual and auditory activity we observed. This combined space is a subspace of the complete neural activity space (Fig. 5a, see details in the Supplementary Text). The animal is required to compute a decision based on these stimuli. The decision will be characterized by an activity vector **d**. The transformation of the activity from sensory stimuli to the decision can therefore be described by the transformation **d** = F(**v** + **a**): F projects the stimulus subspace to the decision subspace. In incongruent trials the animal is expected to learn to arrive at different decisions for identical stimuli such that the transformation F that is implemented by a network of neurons is invariant across contexts. Therefore, contextual

selection can only work if the stimulus subspace has additional modulating activity. We describe this modulatory activity as an additional component in the activity space, denoted by **m**. Thus, with an additive modulatory signal, the decision is actually acting on the joint activities d = F(**v** + **a** + **m**) (Fig. 5a). In order to remove dependency of the output on the irrelevant modality the modulatory signal shall either (i) reduce or remove the activity in that modality, e.g., in the visual context suppression of **a** requires **m** ∈ A, a vector in the auditory subspace pointing to the opposite direction of **a** (Fig. 5a), or (ii) enhance the relevant subspace, with: **m** ∈ V in the visual context with the same vector direction that of **v**. To let the modulation depend on context, this modulatory signal, **m** can be formulated as applying a mapping, **M** to context, **c**; in matrix form: **m**(**c**) = **Mc**. Intuitively, (i) suggests that **m** is suppressing **a** in the visual context and vice versa in the other

context. Critically, the only transformation that fulfills this requirement is cross inhibition of the two stimulus subspaces, corresponding to a matrix with a qualitative block structure of $\mathbf{M} = \begin{pmatrix} 0 & -1 \\ -1 & 0 \end{pmatrix}$ (Fig. 5a). We provide formal proof in Proposition 1 and Corollary 1 (Supplementary Text). Less formally, we argue that subspaces of neuronal activity that are selective to a particular stimulus modality will display strong modulation with changes in context. Given a subpopulation of neurons representing one stimulus modality and another representing the other modality, the modulation is achieved through the mapping $\mathbf{M}$ from the context-representing subspace, $\mathbf{c}$, to the subpopulations representing the stimuli. The scale of $\mathbf{M}$ sets the magnitude of suppression of the irrelevant modality. The elements of $\mathbf{M}$ are the learned suppression levels required for the task, so that suppression overcomes the irrelevant activity. Note, that when computing a decision through applying the mapping $\mathbf{M}$, enhancement of the relevant stimulus modality or suppression of the irrelevant one are equivalent (Supplementary Text). The enhancement can be achieved by having positive diagonal elements, with the qualitative block structure of $\mathbf{M} = \begin{pmatrix} 1 & 0 \\ 0 & 1 \end{pmatrix}$. The combination of enhancement and inhibition yields a simplified schema of the context-dependent modulation of a pair of subpopulations (Fig. 5b). So far, $\mathbf{M}$ was considered to perform a mapping across subspaces, i.e., context was delivered as an input to the stimulus subspace. An alternative scenario can be identified when the context is represented within the same subspace as the stimuli (Supplementary Text). In this case $\mathbf{M}$ is implemented as mutual inhibition between subspaces representing the two stimulus modalities (Fig. 5c). These subspaces do not necessarily correspond to disjunct populations of neurons, and may feature mixed-selectivity neurons that are modulated by both stimulus modalities. For a formal treatment of this more general setting see Corollary 2 (Supplementary Text). The analytical derivation can be further extended to additional important directions: non-orthogonal stimulus subspaces; biologically plausible response nonlinearities; and sequential processing. Sequential processing is an important extension, as recurrent circuitries, characteristic of higher-order brain areas, perform sequential computations and are capable of approximating F as the activity of a neuron population unfolds in time (see Supplementary Text). Most importantly, we conclude that when connection weights are fixed, even under the general conditions of mixed-selectivity neurons and sequential processing of the input, the computation of context-dependent selection of the relevant stimulus has to act on the stimulus subspaces, enhancing the relevant and inhibiting the irrelevant stimulus subspace.

The above argument indicates that context-gated suppression is a potent mechanism to perform the set-shifting task. We next investigated if this mechanism corresponds to the computations that are implemented in our RNN. In our RNN, the recurrent connections have to perform all the computations necessary to reach a decision from the auditory and visual inputs: since context $\mathbf{c}$ is represented in the RNN, it is transformed by recurrent dynamics into modulation $\mathbf{m}$, which in turn is integrated with the stimulus representation $\mathbf{v} + \mathbf{a}$. First, we examined if the trained RNN displays the mutual inhibition pattern required for suppression of the irrelevant modality. We investigated the relationship between the input connections to and the recurrent connections within the hidden layer in RNNs. We therefore calculated a modality index by determining the difference between the strength of the input weights conveying the visual and auditory inputs for each hidden unit. We sorted the hidden neurons by the modality index, so that the first neurons receive strong visual and weak auditory input, while the last neurons weak visual and strong auditory input. We investigated the network structure by displaying the connection matrix for the recurrent layer of the RNN. The connection matrix is

largely unstructured but revealed patterns reminiscent of the transformation M, incorporated into F unfolding in time: diagonal elements showed positive feedback, and far ends of the modality index spectrum revealed traces of inhibition (Fig. 5d). The mutual inhibition structure can be investigated more thoroughly by assessing the effect of iterative processing of the incoming information. The iterative effect was established by calculating increasingly higher powers of the connection matrix, which describes the dynamics of the RNN unfolding in time. To reveal how inputs are transformed by the RNN by the time decisions are made, we plotted up to the 6th power of the recurrent connection matrix, linearly simulating repeated multiplication on the hidden states. We observed a strengthening of the positive self-feedback and mutual inhibition between disjoint block matrices (Fig. 5e). Taken these together, the structure of the lateral connections is characterized by a simple block matrix structure $\begin{pmatrix} 1 & -1 \\ -1 & 1 \end{pmatrix}$.

Next, we sought to establish a signature of inhibition-based manipulation of stimulus representations. The mutual inhibition between the two stimulus modality-specific neuron groups causes each group to be less active in their irrelevant context after suppression. Hence the firing rate differences between activities in the two contexts, i.e., the context representation, will at least partly originate from the context-gated suppression itself. The two computations, the slower context inference and the faster irrelevant suppression, are tied together during active attentional modulation. This coupling can be tested in the trained RNN by establishing the relationship between the modality specificity and contribution to context decoding of individual neurons. We constructed a context index for individual neurons by taking the difference in the activity of each cell between the two contexts. We found significant correlation between the context index and the modality rank of the neurons when the neurons of the hidden layer of the RNN were sorted according to the modality index (Fig. 5f, $n = 30*88 = 2640$, $r = -0.97$, $t = -18$, $P < 10^{-17}$). As context is a variable that informs the animal about the stimulus/outcome contingencies in the upcoming trial, therefore the correlation between the sensitivity to stimulus modalities and modulation by context might reveal an important property of context representation of the animal performing the set-shifting task. Note, that this correlation between the modalities and context does not contradict that stimulus and context representations reside in orthogonal subspaces. The stimulus subspace is characterized by the optimal differentiating vector between the "go" and "no go" instances of the stimulus, as calculated by the stimulus decoder DV. In contrast the difference between modalities signify either visual or audio-related activity in their respective subspaces.

We tested if the context representation in mice abodes the requirement of context-gated suppression we identified in RNN, i.e., neurons displaying higher modulation by stimulus modality changes will be more intensely modulated by changes in context. To explore the specificity of the prediction to ACC, we analyzed neurons recorded in both ACC and V1. For the modality index, we calculated the difference between average responses to visual stimulus and to auditory stimulus. In order to compare neuron responses to each stimulus without interference from the other stimulus modality, we used single modality trials, not part of the complex task, but recorded in the same session. Similar to the RNN analysis, we sorted the neurons according to their modality index. As cells typically encoded either the "go" or the "no-go" signal, modality and context indices were calculated and sorted separately for the two signals, and then combined. The context index was calculated for individual neurons based on the difference of responses in the two task contexts (see "Methods" for details). We found a clear correlation between the context index and neuron modality rank in mice in ACC ($r = -0.40$, $n = 122$, $t = -4.84$, $P < 10^{-5}$, Fig. 5h), but not in V1 ($r = -0.06$, $n = 196$, $t = -0.83$, $P = 0.41$, Fig. 5g). In

order to explore the relation between context and modality representations more generally, we repeated this analysis in short time periods at different stages of the trial. We used −1.0 to −0.25 s period before stimulus start ("pre"), so that the context index contained memory of inferred context between trials, but avoided contamination with stimuli-related activity, and excluded contextual differences due to the suppression of the irrelevant modality. We compared "pre" correlations with that around stimulus start (from 0 s to 0.75 s from stimulus start, "start"), and at decision-making (from 1.75 s to 2.5 s after stimulus start, "dec"). We found a similar correlation at all intervals ($n = 122$, $r$ values −0.28, −0.21, −0.30, t statistics −3.19, −2.31, −3.43, $P$ values 0.002, 0.023, 0.001 for "pre", "start", and "dec", respectively, Fig. 5i), indicating that context information is related to modality representation even in the absence of stimulus and suppression.

In summary, identifying subspaces of stimuli and context within both recorded ACC neurons and the hidden layer of the RNN model revealed the same gated mutual feedback we proved theoretically. The effect of this mutual feedback apparently results in a particular correlated structure of cell responses: Cells that respond more selectively to a stimulus modality tend to be more active in the context where that modality is relevant. This essentially means that the representation of the attention cue, context, and the selection target, modality, is largely overlapping in the same neural population. In this configuration, downstream decision-making centers can dynamically read out the abstracted relevant stimulus representation from the total stimulus subspace without changing the readout weights.

## Discussion

We examined the computations required to establish the relevance of incoming stimuli and the way this computation is reflected in the activity of neurons recorded in ACC. In a changing environment where relevance of stimuli can change, we argue that resolving conflicts between concurrently presented stimuli is key for effective performance. We found that suppression of irrelevant stimuli was a signature of conflict resolution in ACC as suppression prevalently occurred when behavior also reflected that the irrelevant stimulus was successfully ignored. We identified analogous suppression in an RNN model trained to perform the set-shifting task. We proposed that context-gated suppression is a critical computational mechanism for dynamically identifying relevant stimuli from a wider set and we found that the geometry of the representation in the ACC efficiently supports context-gated suppression. We analytically proved that recurrent weights of the trained RNN achieve enhancement of current context and suppression of irrelevant context through mutual feedback. Context-dependent suppression suggested a relationship between the representation of stimuli and that of context, which we demonstrated in the RNN. Assuming context-dependent suppression occurring in higher-order areas, such as ACC, we expected this relationship to be present in ACC, but absent in V1. Our ACC recordings confirmed that similar to RNN, modality and context indices are correlated and this relationship holds not only during stimulus suppression but also in the intertrial interval and early in trials. However, we did not find evidence for suppression or the correlation between stimulus modality and context indices in V1. Thus, the proposed signature of context-dependent suppression is specific to ACC. In summary, our results demonstrate that a context-gated attention mechanism can dynamically resolve instruction conflict during the presence of a distractor.

Our recurrent network model provides important insights into the way relevant stimuli can be identified in a rich environment. The mutual feedback structure identified in the lateral weights of a set-shifting task-trained RNN supports the mechanism that amplifies the currently relevant modality and suppresses the currently irrelevant modality. This architecture in turn supports the maintenance of the current context. Thus, the synaptic weights of the network have a dual role: they both contribute to the modulation of the total stimulus subspace, i.e., the modality targeted by attention, but also contribute to the maintenance of context. This dual role helps understanding the relationship we demonstrated between the context and modality indices of individual neurons.

The activity representing the contextual cue and the target modality of the attentional selection mechanisms are highly correlated; a prominent pattern that emerged from our results. One can argue that this correlation is due to the suppression of the irrelevant modality, as firing patterns after suppression are clearly different in the two contexts with larger activity in the relevant stimulus subspaces. However, we showed that the correlation between modality specificity of neurons and the context representation is present not only during suppression but also prior to stimulus onset. We showed that the cells responsible for the representation of the non-suppressed modality are the ones generally more active during that context. Note, that the trained RNN is both capable of maintaining a constant representation of the incoming stimuli and of representing them in a context-gated manner, with irrelevant stimulus suppressed. This is also characteristic of the representation identified in ACC as evidenced by the activity in the first 500-ms of stimulus presentation where even the irrelevant stimulus is discriminable and full context-driven suppression later in the trial. Similar linear geometry was proposed for attentional selection as a context-dependent stimulus-selector vector[17]. Here we analytically proved that this modulation is the only possible computational mechanism to shift attention on short timescales when weights remain unchanged. Furthermore, we demonstrated that this computation emerges through specific mutually inhibiting input and lateral connections.

The computational motif of mutual inhibition between subspaces representing different modalities was identified (i) analytically, (ii) in RNN models, and (iii) in ACC of behaving mice. The analytic derivation and the recurrent feedback connections of the RNN model explicitly revealed an architecture characterized by mutual inhibition. Modeling studies showed that this connectivity allows switching between distinct dynamical behaviors[18]. Although connection structure between in vivo neurons was not available with our recordings, anatomy of the local circuitry in the cortex still provides insights into the relevant circuitry in ACC. Multiple inhibitory neuron types have been identified that are coupled to pyramidal neurons that control local circuit output and can potentially contribute to lateral inhibition[19]. Such lateral connections are typically part of competing representations in sensory cortices[18,20], in the prefrontal cortex[21], and in the brainstem[22]. Although no such studies are available for ACC, it is likely that similar anatomical motifs contribute to both stabilizing context representation, and upon prediction errors computing the correction of a mismatched context representation in the ACC.

In the ACC, task variables, e.g., stimuli and context, jointly modulated the activities of individual neurons, resulting in a mixed-selectivity representation. Mixed selectivity has been demonstrated in multiple brain areas[4,23,24]. Traditionally, it is the specific selectivity of neurons to certain variables that has been identified[25–27]. While the observed phenomena, such as selective suppression, could be sought in individual neurons, we focused our analyses on population responses. Our analyses, which identify low-dimensional subspaces that accommodate orthogonal representations for task variables, provides an important insight into the relationship between mixed and selective representations: in a rotated coordinate system activity along a particular axis covaries with a single independent variable. As such, activity of a mixed-selectivity neuron belongs to multiple subspaces, while a specific selectivity neuron belongs to an axis-aligned subspace[28,29].

In order to investigate how changed relevance of sensory stimuli is processed in the cortex, we developed a paradigm where the context was not presented to the animals but had to be inferred through trial and error. If the context is cued there is no actual conflict after learning

the task and context-gating function is attributed to the PFC[17]. Beyond conflict resolution, in our experiments ACC might be implicated in reinforcement learning related computations[30]: the performance of our animals saturated below 100% correct responses as they made a number of exploratory trials to optimize their behavior (Fig. 1c). We conclude that when a task is not fully learned and the animal is frequently faced with situations where conflicts arise, the ACC plays a larger role preparing the PFC for the optimal task representation, while post-practice decrease in ACC activity is larger than in the PFC[31,32].

Evidence from our mathematical analysis indicates that unless the animals actually performed this inference they could not successfully perform the task. Earlier research has indicated that components necessary for context inference are represented in the cortical circuitry. PPC has been shown to maintain a history of past environmental information[33], while PFC has been implicated in maintaining previous choices and rewards[13]. Interestingly, the contextual representation appears to be continuous in ACC and V1[4], as well as in our RNN model: the context signal is present during off-stimulus periods. This indicates that the context representation is maintained at a timescale beyond that of a single trial. This is consistent with the idea that storing information in short term memory by vmPFC[34] is more effective than recomputing a constant context on each trial. Nevertheless, the representation should be flexible so that it can be updated if context changes. ACC might be involved in contextualizing goal-oriented sets of stimuli, action and reward upon encountering error feedback[7]. We did measure context-related activity in ACC of mice, thus it could potentially be used to resolve conflict by ACC. Without causal interventions the source of contextual modulation cannot be uniquely identified. However, the specificity of our predictions to ACC provides sound support for the link between context representation and relevant feature selection for decision-making.

We used decoders to identify context-related signals in ACC. Note, that decoders pick up any source of neural activity pattern that differs between the two blocks of a session. Such neural signals might include task-related changes in neural activity or gradual changes independent of the task. Task-unrelated changes can include gradual changes in neural signals across the experimental session that would be reflected as a context signal since the two contexts appeared successively at the beginning and at the end of an experimental session. To control for this potential confound we introduced an analysis that distinguished trials at task boundaries and trials distant from task boundaries. The decoded context signal was not dependent on this choice, which confirmed that no such gradual drift could account for the decoded context. When considering task-related changes, it is not entirely surprising that there appeared to be a contextual signal even when the animals' behavior did not reflect proper task execution, both shown as error trials, and as longer exploratory periods. The signal identified by the context decoder under such unsuccessful conditions might come from different sources: (i) an uncertain representation of context that demands more sampling of the task space, (ii) environmental non-goal-oriented context information, which may come predominantly from PPC[33], and (iii) an asymmetrical task execution, in which one of the contexts is successfully performed but in the other context the absence of goal-oriented knowledge introduces response modulations. We hypothesize that during unsuccessful task execution the outcome history contingencies and prediction errors in ACC can both contribute to a non-goal-oriented context representation[35].

Decoders were used to analyze the suppression of stimulus-related activity when the stimulus was not relevant for the decision in the actual context. As the paradigm combines stimuli, decisions, and contexts, special care is required to eliminate potential confounds coming from correlations between these variables. Suppression of the irrelevant modality coincides with the break-down of correlation between the identity of the presented irrelevant stimuli and the correct decisions the animal makes. Direct disentanglement of stimulus-

related and choice-related activity would require that stimulus-related activity is analyzed when decision is held identical across conditions. Such close control is not permitted by the paradigm, as it would require a sufficient number of error trials. Instead, we introduced a number of controls, which provided strong support for the suppression of irrelevant stimulus modality.

We emphasize that correct execution of the uncued set-shifting task relies on a number of computational steps to be executed, namely context inference and selection of the relevant set. Disruption of any of these seems to result in declining behavioral performance for the animals, with specific signatures that are identifiable in our model as well. Incorrect context inference causes the animal to switch to alternative strategies that are most evident in incongruent trials as: (i) randomly choosing between licking or withholding lick; (ii) reclining to a lick-only strategy; or (iii) making choices according to the rules of the other context, i.e., making decisions based on the distracting stimulus. Our RNN model does not implement complex strategy-exploring algorithms, however strategy "iii" observed in mice was clearly identifiable in the RNN model in the early epochs during learning: In one of the contexts, incongruent trials elicited choices that would have been correct only in the other context. Interestingly, our analyses revealed that mice performed worse in trials where the suppression of irrelevant subspace was not sufficiently strong in ACC. Our model with an emergent simple mutual inhibition reproduces context-specific suppression in a well-trained model. Accordingly, in early learning epochs a strategy reminiscent of "iii" was coinciding with a lack of the suppression of the distractor modality.

The aim of the context-gated suppression is to prepare the correct set of stimuli for a downstream target of ACC, the PFC[5], by selecting the relevant stimulus. Our results demonstrate that in attention-shifting tasks this is achieved by constraining the activity of ACC to a smaller subspace that only accommodates the relevant modality. This computational mechanism has the appeal that a downstream area can read out information relevant for a choice relying on fixed connections that project from the entire stimulus-related subspace[15,16,29], thus resulting in an efficient solution for conflict resolution.

## Methods

### Surgery

All experimental procedures were approved by the University of California, Los Angeles Office for Animal Research Oversight and by the Chancellor's Animal Research Committees. In all, 7–10 weeks old male and female C57Bl6/J mice were anesthetized with isoflurane (3–5% induction, 1.5% maintenance) ten minutes after intraperitoneal injection of a systemic analgesic (carprofen, 5 mg/kg of body weight) and placed in a stereotaxic frame. Mice were kept at 37 °C at all times using a feedback-controlled heating pad (Harvard Apparatus). Pressure points and incision sites were injected with lidocaine (2%), and eyes were protected from desiccation using artificial tear ointment. The surgical site was sterilized with iodine and ethanol. The scalp was incised and removed, and a custom-made lightweight omega-shaped stainless steel head holder was implanted on the skull using Vetbond (3 M) and dental cement (Ortho-Jet, Lang), and a recording chamber was built using dental cement. Mice recovered from surgery and were administered carprofen for 2 days, and were administered amoxicillin (0.25 mg/ml in drinking water) for 7 days. Mice were then water-deprived and trained to perform the behavior (discussed below).

Approximately 24 h before the recording, mice were anesthetized with isoflurane, a small craniotomy (0.5 mm diameter) was made above the right cerebellum and a silver chloride ground wire was implanted within the craniotomy and fixed in place with dental cement. A circular craniotomy (diameter = 1 mm) was performed above the ACC (anterior-posterior 1.8 mm, medio-lateral 0.25 mm). The exposed skull and brain were covered and sealed with a silicone elastomer sealant (Kwik-Sil, WPI). On the day of the recording, the

mouse was placed on the spherical treadmill and headbar fixed to a post. The elastomer sealant was removed and the craniotomy chamber was filled with cortex buffer containing 135 mM NaCl, 5 mM KCl, 5 mM HEPES, 1.8 mM CaCl$_2$ and 1 mM MgCl$_2$.

## Animal training

Following implantation of the headbars, animals recovered over 3 days, and received 10 to 20 min of handling per day, thus habituating the animals to human interaction for 4 days. Animals were then water-deprived, receiving ~1 mL of water per day. During this time, animals were placed on an 8-inch spherical treadmill (Graham Sweet) in the behavioral rig for at least 3 days to habituate to head fixation for 15 min per day. The spherical treadmill was a Styrofoam ball floating on a small cushion of air allowing for full 2D movement (Graham Sweet, England). The animal's weight was measured daily to ensure no more than ~10% weight loss.

Animals were first trained to perform unimodal visual and auditory lick/no-lick (go/no-go) discrimination tasks. Licks are detected by using a lickometer (Coulbourn Instruments). Lick detection, reward delivery and removal, sensory stimulation and logging of stimuli and responses were all coordinated using a custom-built behavioral apparatus driven by National Instruments data acquisition devices (NI MX-6431) controlled by custom-written Matlab code. A 40-cm (diagonal screen size) LCD monitor was placed in the visual field of the mouse at a distance of 30 cm, contralateral to the craniotomy. Visual stimuli were generated and controlled using the Psychophysics Toolbox[36] in Matlab. In the visual discrimination task, drifting sine wave gratings (spatial frequency: 0.04 cycles per degree; drift speed: 2 Hz; contrast: 100%) at 45°, moving upwards, were paired with a water reward. Drifting gratings of the same spatial frequency but at 135° orientation, moving upwards, signaled a reward would not be present, and the animal was trained to withhold licking in response to the stimulus. The intertrial interval was 3 s, except for trials in which the animal had a miss or false alarm, then the intertrial interval was increased to 6.5 s. The animal's behavioral performance was scored as a d' measure, defined as the z-score of the hit rate minus the z-score of the false alarm rate, where z-score is the inverse cumulative function of the normal distribution, converting a probability to units of standard deviation of the standard normal distribution, with smallest allowed margins of 0.01 and 0.99 rates. Once animals reached expert performance (d'>1.7, $P < 0.001$ as compared to chance performance, Monte-Carlo simulation), they were advanced to learning the auditory discrimination task where a low pure tone (5 kHz, 90 dB) indicated that the animal should lick for reward and a high tone (10 kHz, 90 dB) indicated that the animal should withhold licking. The intertrial interval was similarly 3 s, and the intertrial interval was increased to 9 s after misses or false alarms. After animals learned the auditory discrimination task (d'>1.7) they were trained to perform the multimodal attention task. In this phase, animals first performed one block of visual discrimination (30 trials). If their performance was adequate (d'>2.0, correct rejection rate>70%, hit rate >95%) they then performed the visual discrimination task with auditory distractors present (the high or low tones) for 120 trials. Then, after a five-minute break, they performed the auditory discrimination task for 30 trials and if their performance was adequate (d' >2.0, correct rejection rate>70%, hit rate >95%), they performed auditory discrimination with visual distractors present (oriented drifting gratings at 45 or 135°, described previously). During each training day and during the electrophysiological recordings, each trial set started with 30 trials where only visual or auditory stimuli were delivered which signaled whether the animal should base its decisions on the later multimodal trials to visual or auditory stimuli, respectively. Each trial lasted 3 s. When the cue stimulus instructed the animal to lick, water (2 µl) was dispensed two seconds after stimulus onset. No water was dispensed in the no-lick condition. To determine whether the animal responded by licking or not licking, licking was only assessed in the final second of the trial (the response period). If the animal missed a reward, the reward was removed by vacuum at the end of the trial. Animals performed 300–450 trials daily during training. Note that in recording sessions fewer trials were performed. Only one training session was conducted per day with the aim to give the animal all their daily water allotment during training. If animals did not receive their full allotment of water for the day during training, animals were given supplemental water an hour following training. Whether the animal started with the attend-visual or ignore-visual trial set was randomized. Importantly, the monitor was placed in exactly the same way during the auditory discrimination task as it was placed during the visual discrimination task, and a gray screen, which was identical to that during the intertrial interval of the visual discrimination task and isoluminant to the drifting visual cues, was displayed throughout auditory discrimination trials. As a result, the luminance conditions were identical during visual and auditory discrimination trials.

## Behavioral analysis

Performance of the animals was characterized by 21 trials wide sliding window average. Four components were constructed for "go" and "no-go" signals, each having two for congruent and incongruent, and calculated separately in the two contexts. Consistent and exploratory trials were defined as when all four moving averages were above or equal and below chance (0.5), respectively. We estimated the probability of observing at least a number of consistent trials with a simulation of 1,000,000 sessions with the same number of trials as that of the animal with choices randomly sampled from bernoulli distributions where $P$ equaled the mean lick rate in the session. We modeled choices of mice during consistent trial blocks with Bernoulli distributions: $P = 1$ for the contextually correct modality go signal. Opposite models parameterized the distribution with $P = 1$ for auditory signals in visual context and vice versa. We employed a baseline random lick model with a bias so that we set $P$ to the frequency of lick choices. Then a context-aware model was also constructed, with the $P$ equal to the probability of the go instruction, and either a bias, $\beta$, or a lapse, $\lambda$, parameter, so that $P$ equals either $1 + \beta$ and $\beta$, or $1 - \lambda$ and $\lambda$, for go and no-go trials respectively. The $P$ values of all models were clipped between 0.001 and 0.999. Models were compared during consistent incongruent trials by their mean log-likelihoods.

## Electrophysiology

Extracellular multielectrode arrays were manufactured using the same process described previously[37]. Each probe had 2 shanks with 64 electrode contacts (area of each contact 0.02 µm$^2$) on each shank. Each shank was 1.05 mm long and 86 µm at its widest point and tapered to a tip. Contacts were distributed in a hexagonal array geometry with 25 µm vertical spacing and 16–20 µm horizontal spacing), spanning all layers of the cortex. Each shank was separated from the other 400 µm. The electrodes were connected to a headstage (Intan Technologies, RHD2000 128-channel Amplifier Board with two RHD2164 amplifier chips) and the headstage was connected to an Intan RHD2000 Evaluation Board, which sampled each signal at a rate of 25 kHz per channel. Signals were then digitally band-pass-filtered offline (100–3000 Hz) and a background signal subtraction was performed[37]. To ensure synchrony between physiological signals and behavioral epochs, signals relevant to the behavioral task (licking, water delivery, visual/auditory cue characteristics and timing, and locomotion) were recorded in tandem with electrophysiological signals by the same Intan RHD2000 Evaluation Board.

## Microprobe implantation

On the day of the recording, the animal was first handled and then the headbar was attached to head-fix the animal on the spherical treadmill. The elastomer sealant Kwik-Sil was removed and cortex buffer (135 mM NaCl, 5 mM KCl, 5 mM HEPES, 1.8 mM CaCl$_2$ and 1 mM MgCl$_2$)

was immediately placed on top of the craniotomy in order to keep the exposed brain moist. The mouse skull was then stereotaxically aligned, and the silicon microprobe coated with a fluorescent dye (DiI, Invitrogen), was stereotaxically lowered using a micromanipulator into the ACC to a depth of 0.85 mm. This process was monitored using a surgical microscope (Zeiss STEMI 2000). Once inserted, the probe was allowed to settle among the brain tissue for 1 h. Recordings of multiple single-unit firing activities were performed during task engagement (approximately 1 h). After the recording, the animal was anaesthetized, sacrificed, and its brain was extracted for probe confirmation.

### Single-unit activities (SUA)
Spike sorting was performed by Kilosort 2[38], and then manually curated in phy2 using Matlab and Python yielding single-unit activities. Standard consistency criteria were employed for autocorrelograms, interspike interval histograms, waveforms, cluster split and merge, maximal amplitude electrode locations, low false positive or missed spikes and also stable feature projections throughout the recording session. Highly similar clusters were merged manually if crosscorrelation revealed identical refractory periods and if interspike interval histograms and feature distributions matched to provide a resulting unit without drift signs. Clusters were split when PCA feature space and interspike interval histogram showed mixtures of stationary distributions and kept when the cross-correlograms improved.

### Exclusion criteria to control for drift
Although the kilosort algorithm is rather effective to follow drifting units, some more distant units from the electrode shaft was still necessary to curate. We took great care that only units showing no statistically identifiable drift in firing responses were included in the analysis. For this, signal-to-noise ratios of spike events were tracked during the parts of the recording session when the animal performed the task. For any given unit, drift was identified using a set of criteria on a long timescale of the entire session. Specifically, we assessed the quality of the unit based on (A) unit separability of the unit in the PCA feature space from other units, (B) stationarity of signal-to-noise ratio, where noise is the background activity: the spikes of the remaining units projected onto the PCA features, and (C) stationarity of firing rate. We excluded units that did not meet our criteria.

### Spike counts
Spike counts were calculated in 10-ms sliding windows, then Gaussian-smoothed ($\sigma = 100$ ms), approximating single trial instantaneous firing rates (IFR). For decoders IFRs were transformed to z-scores, with mean calculated from prestimulus (from −1500 to 0 ms) time-averaged baseline activities, while standard deviation was calculated for the whole trial. IFRs allowed for grouped trial comparison of mean activity of neurons: differences between group means were calculated as 2 s.e.m. bands around the mean corresponding to 0.05 confidence level.

### Geometry of neural representations
We regarded neural activity as a time-varying vector of baseline-standardized IFRs. The components constitute a basis for the population activity vector space, the possible activity patterns of the neuron population. Classification of trials between values of task variables was trained with logistic regression. Test results were shown as tenfold cross-validation mean over folds, unless otherwise noted. Context decoders were also performed for a non-standard fixed leave-n-out $n = 20$ (10–10 per context) training scheme, with twofold test predictions over 5–5 trials per context. At each time point, we trained separate decoders in standard analyses. The decision vector (DV), **d**, is the normal vector of the boundary hyperplane along which the most difference between classes can be found in trial to trial variation. Angles between decoders DVs were calculated as $\gamma_{12} = \arccos \mathbf{d}_1 \cdot \mathbf{d}_2$,

normalized. DVs of multiple decoders with the same underlying neural basis define low-dimensional subspaces of task-relevant activity. Two-dimensional subspace of two decoders was visualized by the first DV as the first coordinate, and the second DV projected with QR decomposition to the closest orthogonal line to the first DV as second coordinate. IFR time courses, **x**, were projected onto normalized DVs, **d**, by the dot product $\mathbf{y} = \mathbf{d} \cdot \mathbf{x}$. Multiple time points were averaged over for decoder weights, $\mathbf{d} = \langle \mathbf{d}(t) \rangle_t$, and activity from different time ranges were also projected across-time established projections: $\mathbf{y}(t) = \mathbf{d} \cdot \mathbf{x}(t)$. In addition, we examined the stability of representations by projecting activity from one time-period within the trial onto DVs calculated in another time-period. Effective chance level was averaged over 40 independent decoder cross-validation accuracy distributions, with fully randomized trial labels from a Bernoulli ($P = 0.5$) distribution. The threshold consisted of s.e.m. from 40 random samples plus from CVs, with one-sided confidence level. CV-averaged decoder accuracy time points were block-averaged (see below), and the resulting time points were compared for single mice or all mice as distributions over trial time courses with t statistics with the null-hypothesis test of identical means.

### Block averaging time points
We calculated the autocorrelation function of predicted decoder accuracies at various time lags with a resolution of 10 ms. We used the lag = 0.6 s where autocorrelation diminished to 0 as input to block averaging trial time courses for statistical comparisons.

### Predicting neural activity from task variables
We calculated the variance of single neurons predictively explained using a tenfold cross-validated linear regression. Neural activity from smoothed IFR were regressed from combinations of one-hot encoded binary task variable predictors. We used both visual and auditory stimulus identity, and choice as single predictor systems, and also stimulus + choice as two-predictor systems. We compared the variance explained, $R^2$, by choice as the single predictor to the double predictor regression of choice and a single stimulus modality. We then similarly compared $R^2$ of double predictors from the relevant stimulus + choice vs. irrelevant stimulus + choice in each context in consistent trials only. Neuronal activity predictions deemed not reaching a fit, if the cross-validated predictive $R^2 = 1−$residual sum of squares/total sum of squares, was lower than 0. Time points where none of the neurons were valid were omitted from further analysis, we then selected for each time point over the $R^2$-s the best predictable neurons. For display purposes only missing timepoints were linearly interpolated, then we smoothed with a 31-width moving average window. Time points were block-averaged at every 0.6 s window. Missing timepoints excluded in any of the models excluded for comparisons between models. The difference between explained variances were tested against the null hypothesis that the mean of the distribution of block-averaged time point-wise difference was 0 (one-sample $t$ test).

### Recurrent Neural Network Model
A standard recurrent neural network (RNN) model without gates was built to process the same stimuli and trial structure the mice were tasked to: Visual and auditory stimuli were either "go" or "no go", and a decision was to be made during stimulus presentation. Context determined which modality to base the decision on. The decision was compared to the contextual target, and either rewarded or punished for success or failure respectively. The RNN was governed by the following sequential (recursive) update equations:

$$\mathbf{h}_t = f(\mathbf{U}\mathbf{x}_t + \mathbf{V}\mathbf{r}_t^* + \mathbf{F}\mathbf{h}_{t-1}), \tag{1}$$

$$\mathbf{d}_t = f(\mathbf{D}\mathbf{h}_t), \tag{2}$$

where **x** is the multidimensional combined audio and visual input, **r\*** is a maintained reward variable for the outcome in the previous trial (not the previous timestep), **h** is the hidden layer activity vector variable, **d** is the output vector variable, with $t$ and $t$-$1$ timesteps indices. **F** is the recurrent connections for the hidden layer, while **U, V**, and **D** are the stimulus, reward input map weights, and the output map weights respectively. The element-wise non-linearity was chosen $f = tanh$, as all variables were one-hot encoded: E.g. **x** = (1, 0, 0, 1) is combined from a (1,0), "go", visual- and (0,1), "no go", auditory-valued vector. The discrete decision was constructed by round(-softmax(**d**)), i.e., the component with the larger value was taken as the winner between the two components. The reward was calculated as **r\*** = (1,0) for success, when the decision equalled the target, and **r\*** = (0,1) for error trials.

For a single trial, a time resolution of 15 time points was chosen: a 9 point long stimulus presentation with 3 pre and 3 poststimulus points. As opposed to the mouse experiment, we omitted timeout on errors. The network had to decide on the response during the stimulus presentation at time point 7 from stimulus onset, and after the 7th point received reward or punishment until the next trial's stimulus began. We found this time resolution as optimal: Oscillations were typically not present in the hidden activity, but the time course was detailed enough to observe dynamical changes.

A single data point consisted of 5 subsequent trials, each with 15 time points, from the same context in a sequence: 4 trials as a batch of trials during which the model had the opportunity to infer context, and the final trial where the loss function was calculated from the last decision as input to gradients for backpropagation. We generated data combinatorially for 2 choices by 2 modalities in 5-trial sequences, reaching 1024 different sequences total per context. The entire 2048 long dataset of 5-trials sequences was used as training. We did not perform cross-validation for the following reasons: (i) each sequence was unique in the dataset, (ii) each sequence that contained at least one incongruent trial was successfully completed; in contrast there were only cca. 3% (32 + 32) of genuinely unsolvable trials: a sequence type consisting of 4 congruent and 1 final incongruent trial, (iii) there was no additional stochasticity in the data to reduce the role of deterministic context inference.

We assessed the network performance for the number of hidden neurons from 3 to 100, and found that the smaller the network size, the more random initialization determines the learning capacity of the network within reasonable training iterations. We settled at the minimum feasible hyperparameter set: hidden layer size 30, with 5000 training epochs and used the ADAM optimizer with initial learning rate at $10^{-4}$. We trained 100 models with different, fixed random generator seeds. Trained models are available from the link in "Data availability".

### Analysis of the RNN

Linear decoders for task variables were constructed with tenfold stratified cross-validation, the accuracy estimate from each fold averaged, and the s.e.m. calculated. Decoders were calculated at each time point throughout the 5-trial sequence, but some analysis used only fewer time points. Fraction-correct estimates for the network outputs were calculated at each decision requirement point for each of the five trials in the sequence. Hidden layer neural activity trial averages were constructed for stimuli variables. Decoders, fraction-correct estimates, and trial averages were also made available to be calculated from designated subsets of trials, such as incongruent trials, or sequences of specific patterns. Accordingly, angles between stimulus representations were calculated by decoding visual or auditory stimulus from the hidden activity at the first time point of stimulus presentation, so that recurrent connections would not influence the stimulus input. The stimulus DV coefficients are equivalent to the stimulus input weights, denoting stimulus representing subspaces. The angles were similarly calculated as for mice:

$\gamma_{1,2}$ = arccos $d^{(1)}d^{(2)}$, where $d^{(k)}$ is the normalized subspace DVs for $k$th task variable. Projections to stimulus subspaces was similarly calculated to mice: u = **P a**, where u is the one-dimensional projected activity, **a**, of the representative of a task variable, **P** = $d^{(k)}$ is the normalized projection operator as a row vector of the DV in the coordinate system of the projection.

### Task variable representation indices

The mouse modality index was constructed from visual and auditory decoder coefficients: $DV_v^{(i)} - DV_a^{(i)}$, for the $i$th cell, where $DV_s$ is the s (either visual or auditory) stimulus decision vector coefficients. Positive values were assigned to the largest positive value (for go signals class label) of the visual sensitivity, and negative values for auditory responsive neurons. The more mixed a cell's response is, including low response to both stimuli, the closer the index is to the 0 value. The 'no-go' signals were reversely calculated and added together with the "go" signal index. Decoders for stimuli were calculated from early single modality trials in a context block, before the complex trials when the distractor stimulus was also present. This avoided contaminating the neural activity with correlated (congruent trials) and anticorrelated (incongruent trials) activity in the irrelevant stimulus subspace. The RNN modality index was similarly calculated, but DVs were calculated from the first time point of stimulus presentation, and equivalently from the stimulus input weights. The context index was constructed by $x_v^{(i)} - x_a^{(i)}$, for the $i$th cell, where $x_c$ is the mean activity of the cell from the c (either visual or auditory) context averaged over trials and respective time points. Specifically, only congruent trials were included in calculating the context index, as it was important to avoid trivial firing pattern differences due to the reversal between contexts of the relevant "go" and irrelevant "no-go" signals, or vice versa, in incongruent trials. Equivalently context decoder coefficients yielded numerically very similar index values. Ordering single-neuron index values by the descending magnitude for neurons of the other index yielded negative correlation between modality order and context index. For mice, very large and very small neuron indices were discarded for mice with higher number of neurons available so that correlations use similar ordering proximities; this procedure did not yield quantitatively different correlations than without this trimming. Correlating the indices themselves without ordering yielded quantitatively similar r and p values. For convenient display purposes we chose the ordering based trimmed plots.

### Reporting summary

Further information on research design is available in the Nature Portfolio Reporting Summary linked to this article.

### Data availability

Behavior data, spike sorted electrophysiological data, and pretrained RNN models are available in the Zenodo database: https://zenodo.org/record/8379272 (https://doi.org/10.5281/zenodo.8379271). Source data are provided with this paper.

### Code availability

Analysis and model source code is available in the following github repository: https://github.com/CSNLWigner/mouse-acc-rnn-contextgatedattention.

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

## Acknowledgements

P.G. and G.O. were supported by a grant from the Human Frontiers Science Program, P.G. was supported by grants 1R01MH105427, R01NS099137, 1P50HD103557, G.O. was supported by the the European Union project RRF-2.3.1-21-2022-00004 within the framework of the Artificial Intelligence National Laboratory. This research was supported in part by the National Science Foundation under Grant No. NSF PHY-1748958 (G.O.).

## Author contributions

P.O.P., M.E., and P.G. designed and optimized the behavior, and designed all experiments. D.T., M.E., and K.S. trained animals and performed the recordings. D.T., M.V.M., K.S., and M.A.H. curated the data. M.A.H. and G.O. designed the analysis. M.A.H., Z.S., and A.A. performed the analysis with input from G.O. M.A.H., Z.S., A.A. developed the RNN and performed the analysis. M.A.H., Z.S., G.O., and P.G. wrote the manuscript.

## Funding

## Competing interests

The authors declare no competing interests.
