## [Peer Review File · Nature Communications]

Shifts in attention drive context-dependent subspace encoding in anterior cingulate cortex during decision makingREVIEWER COMMENTS

Reviewer #1 (Remarks to the Author):

The manuscript from Hajnal et al investigates the effects of attention on population activity in the anterior cingulate cortex (ACC) of mice performing an un-cued, attention set-shifting task. Specifically, the central goal of this work is to test whether ACC neurons enhance features that are relevant for the performance of a task and suppress those that are irrelevant. The authors rely on electrophysiological recordings in ACC of behaving mice, computational approaches to analyze the data and machine-learning/RNN to model the data.

The authors find that in segments of sessions in which mice perform correctly, activity from ACC neurons encodes best for relevant features compared to irrelevant ones. Furthermore they find that ACC neurons encode task variables. A RNN model was used to reproduce the experimentally observed behavioral performance and neural activity. The model revealed a dependence between context and stimuli representations, which was verified experimentally in ACC. Recordings in V1 showed very different patterns of activity.

This is potentially an interesting manuscript, showing significant results on how ACC (but not V1) functions in a set-shifting task where relevant and irrelevant stimuli are used to differentially instruct behavior in distinct contexts. The use of RNNs is elegant and provides a context for testing the relationship between context-dependence and stimulus responses.

Despite these strengths, the manuscript present major weaknesses that should be addressed.

1) The experimental procedures are not sufficiently described. The authors refer to a previous publication (Hajnal et al 2021) for more details, however all the details necessary for interpreting the results should be provided in the manuscript. For instance, it is not clear how many blocks are performed in each session (two?), how many training and testing sessions are performed for each animal (one?). How long does the training last and how does the performance change across days? How many units are recorded in total for V1 and ACC and for each animal? Does each animal contribute to V1 and ACC recordings? If different animals contribute to V1 and ACC recordings, how does their overall performance differ? Where in ACC did the authors record? Can they provide histological reconstruction of electrode tracks? Are the V1 data from the published V1 dataset? Without answers to these questions (and these are just some examples of the kind of experimental details that should be addressed) it is challenging to rigorously assess the experimental data.

2) The behavioral analysis is quite unclear and does not demonstrate that mice consistently learn the task. As far as this reviewer can understand, mice are probed in sessions consisting of two (?) blocks of 50-150 trials, each preceded by 30 priming trials (for a total of 300-450 trials). Within each session the authors select sub-blocks in which performance exceeds 50%. They call the successful sub-blocks "consistent trials blocks". According to Figure 1D top, the total number of consistent trials in the "consistent trials blocks" is quite small, up to only ~30 trials per session. What is the percentage of consistent trials within the entire session? Also, the authors provide "fraction correct" data only for the trials in the "consistent trials blocks" (Figure 1D second from top). The value is obviously high (as the famous Anchorman quote goes "Sixty percent of the time, it works every time"), but the "fraction correct" value would be more appropriately calculated on the total number of trials in a session. The authors should perform more rigorous analyses of their behavior and exclude animals in which only a fraction of trials shows the correct desired behavior.

3) ACC, neural activity differentiates between relevant vs irrelevant features and consistent vs exploratory blocks. To what extent this difference (as well as the encoding of task variables) can be ascribed to different movements in the distinct trials? The authors should address this issue by including an analysis of licking behavior.

4) The writing is very technical and rich in jargon – particularly for the introduction and discussion. The authors are encouraged to simplify and clarify the text.

5) Minor: the reference Seo and Li has the wrong year (2017 in the text vs 2007 in the ref).

Reviewer #2 (Remarks to the Author):

The authors present an empirical dataset from electrophysiologically recorded mice undergoing an auditory/visual attentional set-shifting task with within-session set-shifting. The obtained data suggests that the anterior cingulate cortex (ACC) orthogonally encodes the local relevance of a sensory stimulus to predict reward and the global relevance of the modality (auditory vs visual) that currently contains reward-related information (context) (Fig. 2, 3). This differs from representations of the same visual cues in visual cortex that are not modulated by stimulus relevance (Fig. 2). Using extensive modeling with a recurrent neural network, the authors claim that ACC performs a suppression of locally task-irrelevant (non-informative) stimuli (or an equivalent enhancement of informative stimuli) and thereby gate or control selective attention (focus on the predictive cue), converting stimulus presentation from a sensory to a relevance-based mode (Fig. 4, 5): "Cells that respond more selectively to a stimulus modality tend to be more active in the context where that modality is relevant. This essentially means that the representation of the attention cue, context, and the selection target, modality, is largely overlapping in the same neural population."

Both, the experimental data and its decoding analysis as well as its extended model-based analysis are timely and very important, and appear in principle suitable for publication in Nature Communications.

However, as an experimental neuroscientist (who is not sufficiently adapted to judge the modelling presented in the Appendix and Figure 5), my main concern is with the necessity of the stated implications of the results. Why are the recordings from ACC consistent with the presented model and not with alternative models? To what extent can the role of relevance-based stimulus suppression really be localized to ACC (i.e. what excludes the possibility, that ACC is simply representing that relevance information inherited from elsewhere)? To what extent are results from the modelling not tautological? - E.g. the conclusion "Therefore neurons in ACC that are selectively engaged in one of the stimulus modalities tend to display strong modulation with task context." and "We found that for a hidden cell, the stronger the weight mapping to the output cells, the more it behaves as a modality-invariant abstract cell. Thus, it responds to the context-relevant stimulus input with enhanced response, and to the irrelevant stimulus input by a negative response (Fig. 4E)." read like tautologies to me, given the modelling goal. A lot of the argument that the model corresponds to the ACC's reality relies on Fig. 4E (related to the description above) and 5F-G, but the data presented therein is already quite remote from the actual recording and "filtered" through the model. Would it be possible to classify cells according to their Context- vs. stimulus-encoding and show decoding accuracies (as in Fig. 2) for these different populations of neurons? Or any other way to demonstrate that more fine-grained aspects of the ACC responses correspond specifically to that model? Also, to what extent are these representation modalities of individual neurons stable over time?

Minor comments:

Figure 1: the authors rely heavily on a previous publication of theirs to describe the task and basic findings like performance and learning parameters of the mice - it is also quite a big jump to go from panel C to D. Possibly such missing information, illustrating the actual behavioural performance of the mice, could be added here, so that the study can be evaluated as an independent piece of work.

The same applies to response properties of the recorded neurons - jumping straight to decoding accuracies, does not really do justice to an appropriate description of activity changes (e.g. time-locked to stimulus-presentation and choice) of the measured neurons.

In Figure 3, - ("minus") seems to be displayed as "?"

Figure 4F is not explained or cited in the main text, 4C is difficult to understand

Figure 5A is missing in the Figures attached at the end

Response to Reviewers

We thank the reviewers for their thoughtful comments and their constructive suggestions. Based on the comments we have updated the manuscript. Our responses can be found to individual comments below. For easier tracking of the updates, in the manuscript we have highlighted changes with blue.

Reviewer #1

The manuscript from Hajnal et al investigates the effects of attention on population activity in the anterior cingulate cortex (ACC) of mice performing an un-cued, attention set-shifting task. Specifically, the central goal of this work is to test whether ACC neurons enhance features that are relevant for the performance of a task and suppress those that are irrelevant. The authors rely on electrophysiological recordings in ACC of behaving mice, computational approaches to analyze the data and machine-learning/RNN to model the data.

The authors find that in segments of sessions in which mice perform correctly, activity from ACC neurons encodes best for relevant features compared to irrelevant ones. Furthermore they find that ACC neurons encode task variables. A RNN model was used to reproduce the experimentally observed behavioral performance and neural activity. The model revealed a dependence between context and stimuli representations, which was verified experimentally in ACC. Recordings in V1 showed very different patterns of activity.

This is potentially an interesting manuscript, showing significant results on how ACC (but not V1) functions in a set-shifting task where relevant and irrelevant stimuli are used to differentially instruct behavior in distinct contexts. The use of RNNs is elegant and provides a context for testing the relationship between context-dependence and stimulus responses.

We thank the reviewer for their kind words

Despite these strengths, the manuscript presents major weaknesses that should be addressed.

1) The experimental procedures are not sufficiently described. The authors refer to a previous publication (Hajnal et al 2021) for more details, however all the details necessary for interpreting the results should be provided in the manuscript. For instance, it is not clear how many blocks are performed in each session (two?), how many training and testing sessions are performed for each animal (one?). How long does the training last and how does the performance change across days? How many units are recorded in total for V1 and ACC and for each animal? Does each animal contribute to V1 and ACC recordings? If different animals contribute to V1 and ACC recordings, how does their overall performance differ? Where in ACC did the authors record? Can they provide histological reconstruction of electrode tracks? Are the V1 data from the published V1 dataset? Without answers to these questions (and these are just some

examples of the kind of experimental details that should be addressed) it is challenging to rigorously assess the experimental data.

We now clarify that two blocks were performed in each session. We also now indicate that training typically took several weeks and the details of training have been outlined in our previous paper on V1 neural dynamics (Hajnal et al., 2023, Nature Communications). The number of units are now clearly outlined in the paper. We now indicate that different animals contributed to V1 and ACC recordings. The V1 data are from the published V1 dataset. This is now clearly outlined in the paper. Due to technical issues we could not recover the electrode locations from these ACC recordings, but stereotactic coordinates correspond to cg1 domain of ACC.

2) The behavioral analysis is quite unclear and does not demonstrate that mice consistently learn the task. As far as this reviewer can understand, mice are probed in sessions consisting of two (?) blocks of 50-150 trials, each preceded by 30 priming trials (for a total of 300-450 trials).

We have updated the text to clarify the schedule of a recording session. Importantly, the typical length of a block of trials is shorter than those quoted by the Reviewer. The range of dual-modality block lengths spanned 40-80 trials, therefore the lengths of sessions ranged 160-200 trials. Note that the methods section quoted 300-450 trials per session during training, but the session length was short for recording sessions. We now include a figure that clearly provides this information (Suppl. Fig. 1A-D) and have clarified this in the Methods section too (pg. 5, lines 159-168; pg. 28, lines 1119-1120).

Within each session the authors select sub-blocks in which performance exceeds 50%. They call the successful sub-blocks “consistent trials blocks”. According to Figure 1D top, the total number of consistent trials in the “consistent trials blocks” is quite small, up to only ~30 trials per session. What is the percentage of consistent trials within the entire session?

We have added this percentages to the manuscript (pg. 5, lines 173-176), and the detailed performance of individual animals are now plotted in Supplementary Fig. 1.

Also, the authors provide “fraction correct” data only for the trials in the “consistent trials blocks” (Figure 1D second from top). The value is obviously high (as the famous Anchorman quote goes “Sixty percent of the time, it works every time”), but the “fraction correct” value would be more appropriately calculated on the total number of trials in a session.

Since the strategy of the animals is shifting during an experimental block, the fraction correct value over the whole block alone is not providing full insight into the behavior of an animal during the experiment. We observe the animals exploring strategies, including collecting information in uncertain situations, and then arriving at correct response strategies visible in consistent trials. However, we agree that a more rigorous analysis of behavior is important to prove that good performance in clustered consistent trials are not due to chance; it is this analysis that we describe below.

The authors should perform more rigorous analyses of their behavior and exclude animals in which only a fraction of trials shows the correct desired behavior.

We thank the reviewer for highlighting this point. We have extended the behavioral analysis with a statistical analysis of the individual animals' performance. The statistical analysis assesses the performance of the animals in individual blocks against the alternative hypothesis that the consistent blocks occurred by chance through a behavioral strategy that is ignorant to the task context. The analysis demonstrated that consistent blocks are highly significant in five out of eight blocks, and in the remaining three blocks the chance of the observed consistent block is still below 0.05.

These analyses are now included on Figure 1 for each animal and a more extended version of the analysis is included as a supplementary figure (Suppl. Fig. 1E-H). The main text and the methods have been extended accordingly (pg. 5, lines 173-190; pg. 28, lines 1134-1137).

Please note that the analysis of neural activity during suboptimal task performance is an important element of our study, which reveals key properties of neural computations. Therefore elimination of animals from the cohort would significantly limit the scope and impact of the paper.

3) ACC, neural activity differentiates between relevant vs irrelevant features and consistent vs exploratory blocks. To what extent this difference (as well as the encoding of task variables) can be ascribed to different movements in the distinct trials? The authors should address this issue by including an analysis of licking behavior.

We thank the reviewer for recommending decision and movement controls. We performed three additional analyses to address these questions.

First, to control for the potential effect of licking differences between consistent and exploratory behaviors, we tested irrelevant stimulus suppression on lick-only trials. Note that focusing exclusively on no-lick trials was not possible because of the low number of these trials (due to a tendency of the animals to lick, a strategy for gathering rewards even under uncertainty). We compared the time course of stimulus decoder accuracies of these lick-only trials with decoder accuracies established for all trials. We found similar time courses in the two conditions. Note, that the best performing animals are not available for such an analysis due to a lack of a sufficient number of exploratory trials (see illustration of this point in Fig. R1). We included this analysis in Supplementary Figure 3A-D, and added relevant text to the Results (pg. 8, lines 292-303).

Second, we projected stimulus evoked activity from successful 'go' and 'no go' trials to the stimulus representation subspace identified by the early activity (0.25-0.75s) stimulus decoder DVs. This choice prevented direct effects from choice and licking behaviours

from contaminating the analysis. We used the early activity decoder subspace to investigate population trajectories occurring later during the trial. Population activity was projected on the DV subspace and was averaged across trials with the same stimulus. Difference of population trajectories in the two stimulus conditions was calculated and compared in relevant and irrelevant conditions. We found that the difference of population trajectories was significantly more pronounced in 7 out of all 8 blocks, while 1 modality discriminability was reversed during mid trial (Supplementary Figure 3E-H). This analysis highlights that in a subspace that is dominated by stimulus the effect of suppression can be consistently identified. We argue that these analyses illustrate that choice has likely no influence on the suppression of the irrelevant activity late in the trial. Beyond including this analysis in Supplementary Figure 3, we also extended the main text (pg. 8-9, lines 305-321).

Third, we introduced a single-cell analysis to identify contributions of stimulus presentation to neural activity that is separate from the contributions of choice. This single-cell analysis with binary constant variables is naturally more noisy than population analyses, still we found evidence of separate contribution from choice and the visual stimulus (Supplementary Figure 3I-L). We also extended the manuscript (pg. 9, lines 323-335)

Figure R1. Visual (top) and auditory (bottom) stimulus decoder accuracies in the irrelevant context in exploratory trials in licking only (black lines) and no-lick only (red lines) - thus both successful and error - trials. In comparison: the same, but in all trials regardless of licking response (orange lines, same as Supplementary Fig. 2F orange lines). If the overall, either 'go' or 'no go' number of trials were low (< 10), that particular decoding was omitted from the figures.

Note that a good performing context for an animal yields few number of trials in exploratory trials, while subpar performance allows for using more trials to test potential licking effects in this analysis.

4) *The writing is very technical and rich in jargon – particularly for the introduction and discussion. The authors are encouraged to simplify and clarify the text.*

We thank the reviewer for this remark. We have rewritten the introduction and eliminated overly technical and jargon.

5) *Minor: the reference Seo and Li has the wrong year (2017 in the text vs 2007 in the ref).*

We thank the reviewer for pointing this out. We have now corrected the year of publication, 2007.

Reviewer #2

The authors present an empirical dataset from electrophysiologically recorded mice undergoing an auditory/visual attentional set-shifting task with within-session set-shifting. The obtained data suggests that the anterior cingulate cortex (ACC) orthogonally encodes the local relevance of a sensory stimulus to predict reward and the global relevance of the modality (auditory vs visual) that currently contains reward-related information (context) (Fig. 2, 3). This differs from representations of the same visual cues in visual cortex that are not modulated by stimulus relevance (Fig. 2). Using extensive modeling with a recurrent neural network, the authors claim that ACC performs a suppression of locally task-irrelevant (non-informative) stimuli (or an equivalent enhancement of informative stimuli) and thereby gate or control selective attention (focus on the predictive cue), converting stimulus presentation from a sensory to a relevance-based mode (Fig. 4, 5): "Cells that respond more selectively to a stimulus modality tend to be more active in the context where that modality is relevant. This essentially means that the representation of the attention cue, context, and the selection target, modality, is largely overlapping in the same neural population."

Both, the experimental data and its decoding analysis as well as its extended model-based analysis are timely and very important, and appear in principle suitable for publication in Nature Communications.

We thank the Reviewer for their encouraging words.

However, as an experimental neuroscientist (who is not sufficiently adapted to judge the modelling presented in the Appendix and Figure 5), my main concern is with the necessity of the stated implications of the results. Why are the recordings from ACC consistent with the presented model and not with alternative models?

We thank the Reviewer for raising this issue. The central contribution of the paper is to link the way neuronal populations represent task variables and the computations necessary for performing a challenging task, the set-shifting paradigm. We showed that independent features are represented in orthogonal subspaces and presented a mathematical argument that under the constraint of a fixed-weight circuitry this representation there is only one possible way to modulate activity so that the computations for the appropriate action selection can be performed. Specifically, we have shown that representations that rely on orthogonal linear subspaces to accommodate task-relevant variables invoke vector addition and subtraction to identify features relevant for decisions. This link between representations and computations is then tested in simulations and neural recordings, observed as suppression of irrelevant stimulus. As the link between representations and computations was established by the task being performed, we sought to establish if the relationship we identified was specific to ACC and therefore a critical comparison concerned the properties of ACC and that of V1. While the representation of stimuli and that of context in ACC were interdependent, as signified by the correlation between the representation of stimulus modalities and that of context, the context representation in V1 was shown to be qualitatively different, residing in an independent subspace.

We clarified these points in the Discussion (pg. 20 lines 789-795).

To what extent can the role of relevance-based stimulus suppression really be localized to ACC (i.e. what excludes the possibility, that ACC is simply representing that relevance information inherited from elsewhere)?

As our recordings were constrained to ACC and V1, we could not exclude the possibility that the predicted properties were inherited from other brain regions we did not record. However, the specificity of our predictions to ACC provides support for the link between representation and computations being present in ACC. Our results imply that ACC and its efferent targets are modulated by this relevance-based suppression (no matter the exact source), and affect local and downstream computations. We now clarify this point both in the Introduction (pg. 3, lines 102-103) in the Discussion (pg. 21, lines 821-824).

To what extent are results from the modelling not tautological? - E.g. the conclusion "Therefore neurons in ACC that are selectively engaged in one of the stimulus modalities tend to display strong modulation with task context." and "We found that for a hidden cell, the stronger the weight mapping to the output cells, the more it behaves as a modality-invariant abstract cell. Thus, it responds to the context-relevant stimulus input with enhanced response, and to the irrelevant stimulus input by a negative response (Fig. 4E)." read like tautologies to me, given the modelling goal.

We thank the Reviewer for raising this issue. Our analytical treatment revealed that context-dependent suppression is required for efficient task performance. This indicates

that a context-dependent signal can be identified in stimulus modulated neurons, and the theory also provides an intuition that, as the suppression is stronger for neurons more engaged in stimulus representation, the level of contextual modulation depends on the level of strength of the modulation caused by a particular modality. Our RNN simulations served to provide a confirmation to the above theoretical requirement. In our analysis our goal was to demonstrate that the context representation we identify in animals reflects this property. In the original submission we established the strength of the modulation by a particular stimulus modality through the analysis of trials in which only a single stimulus modality was present. This ensured that the effect of stimulus was established in a condition where the context signal was not required for making a decision. However, as the reviewer pointed out, in the original submission the strength of context modulation was solely assessed in the time window where suppression was already present. Inspired by the question of the Reviewer, we extended the analysis of the context signal such that we can prove that the context representation suggested by the theoretical argument can be identified not only during the period when suppression occurs, but also at stimulus onset and in the intertrial interval. This extended analysis highlights that the relationship between stimulus representation and context representation is maintained throughout the trial, thus fulfilling the need to maintain the contextual modulation across trials.

In practice, we repeated the original analysis such that the context index was calculated for the -1.0 s to -0.25 s before stimulus time window (constraining to the intertrial interval) and in the time window immediately after the time when the sensory stimulus reaches ACC (0 s to 0.75 s).

We clarified and updated the text according to the new analyses (pg. 17-18, lines 668-680, 687-696, and 740-757) and also updated Figure 5 accordingly. We have also clarified the quoted text about the properties of the RNN model (pg. 14, lines 507-524; pg. 19, lines 741-744)

A lot of the argument that the model corresponds to the ACC's reality relies on Fig. 4E (related to the description above) and 5F-G, but the data presented therein is already quite remote from the actual recording and "filtered" through the model. Would it be possible classify cells according to their Context- vs. stimulus-encoding and show decoding accuracies (as in Fig. 2) for these different populations of neurons? Or any other way to demonstrate that more fine-grained aspects of the ACC responses correspond specifically to that model?

We thank the reviewer for highlighting this point. We have made modifications to the analyses and text to substantially increase the alignment between the model and the experiment, as well as we have clarified the role of analyses in the RNN that go beyond the analysis we did with the experimental data. Importantly, we introduced three new panels (Fig. 4E-G), which have direct experimental counterparts. Below we summarize the rationale behind the updated Figure 4.

Fig. 4B does not have a direct experimental counterpart. This panel describes network performance and network properties during training. As the experimental analysis is available for a single session, such potential correspondence between ACC properties and behavior are not possible to obtain. However, the panel provides some important insights: 1, similar to the experiments, there is an asymmetry in the learning to perform well in congruent and incongruent trials; 2, the plot highlights that successful learning of incongruent trials is contingent on acquiring a context variable.

Fig. 4C delivers an insight about the RNN that exploits our ability to analyze network behavior on a trial-by-trial basis upon inferring a new context . As there is a single context change in the experimental data, noisy population responses prevent a trial-by-trial analysis of neuronal data, thus the panels delivers a modeling insight instead of a testable population phenomenon.

Fig. 4D has a direct experimental counterpart.

Fig. 4E,F have been newly introduced to analyze the relationships of stimulus, decision, and context.

Fig. 4G has been newly introduced to directly compare the time course of the population activity during trials, and it is a direct analog of the analysis we introduced for the analysis of the experiments.

Fig. 4H provides an insight about the dynamics of a neuron that is detached from direct sensory effects but is close to network output. This type of analysis is not accessible in experiments, instead it shows through the model how ‘abstraction’ emerges as a result of network dynamics.

In summary, existing analyses along with newly introduced analyses highlight that several key aspects of the neural code identified in ACC can also be identified in analogous analyses in the RNN. These include the asymmetry in task performance, presence of a context representation, the representational geometry of the neuron population, the maintenance of past outcomes in the network, and suppression of stimulus representation when the modality of the stimulus is not relevant for the outcome of the trial.

We have updated the description of the figure in the main text to more efficiently convey the parallels (pg. 13-14, lines 488-492, 496-503, and 507-524).

Also, to what extent are these representation modalities of individual neurons stable over time?

Stability of representations within subspaces across trials were observed, with stable mean and variance, which translates into stable individual neurons comprising the weighted components of representation directions.

Although there is representation drift over time within trials on a finer time-scale, the semantic meaning of the representations is constant, e.g. angle between variable representations are preserved, discriminability between stimuli instructions (difference of responses between 'go' and 'no go' trials) are preserved, even when projected onto subspaces established through the analysis of shorter periods (Suppl. Fig. 2B, Suppl. Fig. 3E-H). Some representations remain relatively fixed, for example context changes minimally between qualitatively different time periods, off- early- and suppressed stimulus time periods, as evidenced by correlating with context index calculated from multiple time periods (Fig. 5E-G).

Minor comments:

Figure 1: the authors rely heavily on a previous publication of theirs to describe the task and basic findings like performance and learning parameters of the mice - it is also quite a big jump to go from panel C to D. Possibly such missing information, illustrating the actual behavioural performance of the mice, could be added here, so that the study can be evaluated as an independent piece of work.

The same applies to response properties of the recorded neurons - jumping straight to decoding accuracies, does not really do justice to an appropriate description of activity changes (e.g. time-locked to stimulus-presentation and choice) of the measured neurons.

We have extended the description of the behavioral training (pg. 5, lines 159-168). Also, we have added a new panel to Figure 1 and additional detail is provided through Supplementary Figure 1, which includes separate analyses for individual animals. We have also added additional behavioral measures to the main text (pg. 5, lines 173-190).

We added a raster plot of neural spikes of several trials to Figure 2, and also included example neurons that behave in a way that their activity is different between relevant and irrelevant conditions when in response to stimuli. These examples illustrate a potential relevance-modulatory effect. We improved the discussion of specific and mixed selectivity representations (pg. 20, lines 789-795).

In Figure 3, - ("minus") seems to be displayed as "?"

We thank the Reviewer for pointing this out. This was an error in the pdf renderer, we have recertified this issue.

Figure 4F is not explained or cited in the main text, 4C is difficult to understand

Thank you for highlighting this point, we added the statement and citation to the text for 4F.

Figure 5A is missing in the Figures attached at the end

This was also an error in the pdf pipeline, we now corrected this issue.

Reviewer #3

This manuscript presents data from an interesting behavioural task in which mice have to attend to one of two simultaneously presented auditory and visual cues, performing go-nogo decision making on the attended modality while ignoring the other modality. The correct modality changes within session allowing the authors to examine stimulus related neural related activity in attended and ignored conditions, as well as activity related to the current context and decision. The manuscript presents data from ephys recordings in ACC and V1, with ACC data the primary focus as the V1 data is discussed in a separate paper, along with computational modelling of the attentional selection process. The main experimental claims of the paper are:

- Attended stimuli are more strongly represented in ACC than unattended stimuli, but this is not the case for V1.*
- The current context (which modality is relevant) is coded in ACC throughout the trial.*
- Representations of the two different modality stimuli are in orthogonal subspaces of ACC activity, which are also orthogonal to the subspace encoding the current context.*

These findings are perhaps not massive surprising, but nonetheless a useful contribution to the literature. I do have a couple of technical concerns about the data and analyses that led to them however.

First, my understanding is that the entire ACC recording dataset in this study consists of one session each from 4 mice. This is both a small number of subjects and a small amount of data per subject, so it is important to make clear to the reader both how consistent effects are across the 4 subjects, and how much data there is per subject. Specifically, please ensure that for all analyses you either show individual points for each subject, or plot the analysis separately for each subject in supplementary material. Also, for each subject please show the behaviour from the recording session, and report the number of neurons, and number of trials of each type for that session.

We have extended the manuscript with a supplemental figure to include behavioral performance of all animals (Suppl. Fig. 1). We performed additional behavioral analysis on identifying ‘consistent blocks’, which provides a quantitative insight into the consistency of behavior across experimental blocks and across animals (Fig. 1D). Details on trial numbers and neuron population sizes have also been added to the text (pg. 5, line 173-190; pg. 6, lines 209-226). Along this line, neural activity for individual ACC-recorded animals is now presented as a supplementary figure (Suppl. Fig. 2).

Second, regarding the claim that attended stimuli are more strongly represented in ACC than unattended stimuli (Figure 2), it would be important to verify that this apparent effect is not due

to representation of the go-nogo decision rather than the stimulus itself. Representation of the decision could contribute to the effect because relevant stimuli will be highly correlated with the go-nogo decision but irrelevant stimuli will not, so representation of the decision will yield decodability of the relevant stimulus. One way to address this would be an encoding analysis predicting neural activity using a linear regression (running separate regressions for each neuron, and timepoint in trial), using the auditory stimulus, visual stimulus, decision, and context as predictors. You would train the regression on all trial types and then evaluate the coefficient of partial determination on held out data split by relevant/irrelevant for each modality, to quantify how much variance is uniquely explained by each stimulus that cannot be explained by the decision. Another complementary analysis would be to show the angle between the decision vector for the stimulus decoders and decision decoder as a function of time from stimulus onset (as in figures 3E,F).

We thank the reviewer for pointing this out. We performed two additional control analyses to address this.

To control for the possibility that movement related activity contributes to the observed suppression we repeated the decoding analysis for different modality stimuli such that trials were restricted to lick-only trials. This condition severely limited the number of trials available for training a decoder, and therefore only a limited number of blocks could be taken into account for this analysis. Our analysis confirmed that suppression can be identified in lick-only trials as well. Moreover, while exploratory trials had less suppression than consistent trials, a result identical to when not lick-controlled (Supplementary Fig. 2).

Next, we tested the potential account that early responses are identical in attended and unattended conditions and late responses only differ because of different decisions/outcomes. For this we assessed activity in the one-dimensional subspace that is identified at the earliest appearance of stimulus-related activity. By investigating this subspace throughout the trial we argued that if the difference between attended/unattended condition is a result of decision-related activity, the suppression we identified would be in a subspace different from the subspace where early stimulus induced activity. To test this scenario, we projected population trajectories onto the decision vector of the stimulus decoder established for the 250–750 ms window and assessed the difference between responses to 'go' and 'no go' stimuli in the population activity projected onto this subspace. We found that suppression was consistently present in this subspace across animals for irrelevant stimuli, while not present for relevant stimuli. We added this analysis as a supplemental figure (Supplementary Fig. 3E-H) and updated the text accordingly (page 8-9, lines 305-321).

Finally, we performed the analysis suggested by the Reviewer: we introduced a single-cell analysis to identify contributions of choice to neural activity that is separate from the contributions of stimulus presentation. This single-cell analysis is naturally more noisy than population analyses, still we found evidence of separate contribution

from choice and the visual stimulus (Supplementary Figure 3I-L). We also extended the manuscript (pg. 9, lines 323-335).

The second half of the paper describes two pieces of computational work modelling the experimental data. The first is an RNN trained on a simulated version of the task, while the second are mathematical arguments about how attention should affect stimulus representations. I did not find the RNN simulations very illuminating because the authors did not apply the same analysis approach to the RNN activity as to the neurons, making it difficult to assess whether the RNN solves the task using similar mechanisms to the mice. Specifically, the authors do not attempt to apply the analysis in figure 2 characterising the strength of stimulus representation in attended and unattended conditions, nor the analyses in figure 3D-F characterising the geometry of the representations. I assume this is because, as they state in the text, ‘Stimulus decoders cannot directly be applied to all the hidden units in our shallow RNN because they would selectively pick activity from cells with strong stimulus input’. But this limitation suggests that this modelling approach is not ideal for understanding this experimental data.

We thank the reviewer for highlighting this point. We have made modifications to the analyses and text to substantially increase the alignment between the model and the experiment, as well as we have clarified the role of analyses in the RNN that go beyond the analysis we did with the experimental data. Importantly, we introduced three new panels (Fig. 4E-G), which have direct experimental counterparts. Below we summarize the rationale behind the updated Figure 4.

Fig. 4B does not have a direct experimental counterpart. This panel describes network performance and network properties during training. As the experimental analysis is available for a single session, such potential correspondence between ACC properties and behavior are not possible to obtain. However, the panel provides some important insights: 1, similar to the experiments, there is an asymmetry in the learning to perform well in congruent and incongruent trials; 2, the plot highlights that successful learning of incongruent trials is contingent on acquiring a context variable.

Fig. 4C delivers an insight about the RNN that exploits our ability to analyze network behavior on a trial-by-trial basis upon inferring a new context. As there is a single context change in the experimental data, noisy population responses prevent a trial-by-trial analysis of neuronal data, thus the panels delivers a modeling insight instead of a testable population phenomenon.

Fig. 4D has a direct experimental counterpart.

Fig. 4E,F have been newly introduced to analyze the relationships of stimulus, decision, and context.

Fig. 4G has been newly introduced to directly compare the time course of the population activity during trials, and it is a direct analog of the analysis we introduced for the analysis of the experiments.

Fig. 4H provides an insight about the dynamics of a neuron that is detached from direct sensory effects but is close to network output. This type of analysis is not accessible in experiments, instead it shows through the model how ‘abstraction’ emerges as a result of network dynamics.

In summary, existing analyses along with newly introduced analyses highlight that several key aspects of the neural code identified in ACC can also be identified in analogous analyses in the RNN. These include the asymmetry in task performance, presence of a context representation, the representational geometry of the neuron population, the maintenance of past outcomes in the network, and suppression of stimulus representation when the modality of the stimulus is not relevant for the outcome of the trial.

We have updated the description of the figure in the main text to more efficiently convey the parallels (pg. 13-14, lines 488-492, 496-503, 507-524).

The final section of the paper makes mathematical arguments about how a contextual stimulus should modify stimulus representations to implement selective attention. I found the setup here rather strange, as the process of selectively attending to one of two stimuli based on a context signal inherently requires a non-linear interaction between the context and stimuli, but the authors framing of the problem is linear. The authors finesse this by making the matrix M mapping the context input to the stimulus subspaces depend on the stimuli themselves, but this is inconsistent with their description of M as ‘fixed input connections’ from the context input to the stimulus subspaces. Further, they define M differently at different places in the manuscript, sometimes giving it fixed values independent of the stimulus as $M=[[0,-1],[-1,0]]$ (e.g. in the main text at the bottom of page 11), and in other places making it dependent on the stimulus as $M=[[0,-v],[-a,0]]$ (e.g. at the bottom of page 19). The interpretation of M also appears to shift between sections, as in some places it is characterised as mapping a context input into stimulus subspaces, whereas in other places it interpreted as mutual inhibition between populations encoding the different stimuli, i.e. a mapping from the stimulus-subspaces back onto these same stimulus-subspaces. Given these issues I did not find the arguments in this section clear enough to yield insight into the attention mechanisms explored in the experiment.

We thank the reviewer for pointing out an unclear and somewhat mixed definition of M , the linear mapping from a context signal onto the stimulus subspace, and F the output mapping to the decision space, which in the recurrent extension also contains the lateral mapping within the stimulus subspace. We have clarified this point in the manuscript (pg. 15, lines 572-595; pg. 17, lines 649-652) but also provide here an itemized explanation.

- First we repeat here, as the reviewer assessed, that the proof works for time-scales in which synaptic weights are not changing, e.g. a single session.
- We argue that the function $m(c)$, which depends on the one-hot encoded context vector (which can be identified with $[1,0]$ for visual context, and $[0,1]$ for audio context in a coordinate system that represents c along a single dimension corresponding to a context subspace, C , as described in Corollary 1), indeed can be a linear map onto the stimulus subspace. This linear map can be written in matrix form in the subspace coordinates for C and $A + B$ respectively on the domain and range: M , hence $m(c) = Mc$. The simplest form of this M matrix is either the self-enhancing identity matrix $\begin{pmatrix} 1 & 0 \\ 0 & 1 \end{pmatrix}$ or the mutual inhibition matrix $\begin{pmatrix} 0 & -1 \\ -1 & 0 \end{pmatrix}$: these mappings enhance the activity of the contextually correct space or decrease the contextually irrelevant space, respectively.
- The amount of enhancement or inhibition can be encoded in fixed elements of the M matrix, thus M does not need to depend on the activities in the stimulus subspace: if the network learned the statistics of the stimuli the mean suppression necessary for repeated task performance can be encoded in fixed synaptic weights. Nevertheless, the reviewer is correct that one of the example forms of matrix M in the manuscript depended on the actual stimulus activity values. With this example we intended to illustrate the necessary amount of inhibition to perfectly neutralize the irrelevant subspace activity. We clarified the aim of this example in the manuscript and changed the proof to rely mainly on the argument that the values of M can be thought of as the mean suppression multipliers. This should separate more clearly that learning establishes optimal mapping weights that creates cross-inhibition, and such optimized fixed weights allow sufficient suppression for optimal task execution.
- Here we clarify the role of M and F matrices. We clarified the description of input map M to better reflect that it is not mutual (recurrent/lateral) inhibition from and to the same subspace, but rather just cross-inhibition from a context representation subspace to stimulus subspace. We added an illustration for this to Fig. 5B (also shown here below as Fig. R2), which now has a cross-inhibiting input mapping on the left, and a mutual inhibition on the right (red lines). Both have enhancement as well (green lines), self-propagation for the input mapping (left), and self-enhancement for the lateral mapping (right). We have restructured the text to better reflect this logic. We elaborated on the theoretical requirements for a closed recurrent system where a locally computed context and the stimuli are both present within the total stimulus subspace, thus M will equal F (the domain and range of M is the same $V+A$ subspace, but also notice that F also has to map to D , thus this is a specialized version of the general case in the proposition and the two corollaries) by adding a relevant sentence to Corollary 2, point (v).

Figure R2. Schematics of amplification and cross lateral-inhibition as a map from the context subspace (grey circles) into the total stimulus subspace (black circles) represented by the connection structure of two axis aligned two dimensional systems: two input neurons acting on two output neurons (left). Schematics of mutual feedback in the total stimulus subspace represented by an axis aligned two dimensional system: two neurons with mutual inhibition and positive self-enhancement (right).

Minor issues:

- *When reporting statistical tests please indicate the experimental unit and give more complete reporting of the test stats (e.g. for t-tests it is standard to report the t value and degrees of freedom in addition to the P value).*

We have updated the text manuscript accordingly.

- *Figure 1C: Missing a Y axis label. What do the solid vs dash lines indicate? Does this show a whole session or part of a session? The methods text indicates sessions are ~400 trials long but this plot shows < 100 trials.*

We thank the Reviewer for pointing these out. Recording sessions consisted of approximately 60-120 multimodal trials per context block. The figures have been amended, session and blocks lengths added and visualized for each mouse in Supplementary Fig. 1, and text in methods has been clarified.

- *Figure 2 legend: What does 's.e.m. also containing across animal mean of CV s.e.m.' mean?*

The confidence band of these across-animal decoders consist of the mean accuracy s.e.m. over mice plus the mean over mice of the s.e.m. of decoder accuracies over CVs for each mice. We added clarification to the figure caption.

- *Figure 4C: Why is the fraction correct ~0.9 on the third stimulus presentation when the model should have no information about the correct context as the first two trials were congruent?*

We thank the Reviewer for pointing out this discrepancy. When decision fraction correct curve is calculated over all sequences, there is an increasing probability of encountering incongruent trials until a given trial within the sequence, which we wanted to illustrate on the figure, but accidentally mislabeled it as a subtype of all sequences in the caption; now we changed this to a gray line, and removed the erroneous solid and dashed styles

corresponding to congruency. Note that congruent trials give 0.75 fraction correct alone. We added the decision fraction correct line for the correct sequence type with orange color, so that it corresponds to the magenta context line now. We inserted clarifications in the figure caption.

- Figure 4E: Is this all trials or incongruent trials only?

All trials.

- Top of page 11: What is the justification for saying 'our RNN model reproduces the ... geometry of the representations of the stimuli and the context' when you did not run the analyses used to assess the geometry of representations in the mice on the RNN data?

Thank you for noticing limited evidence presentation. We now added to Fig 4. the same analysis for RNN evidencing that the representation geometry is indeed the same as for mouse brains.

- I commend the authors for making their data and code publically available, well done.

Thank you.

REVIEWER COMMENTS

Reviewer #1 (Remarks to the Author):

The authors have extensively revised the manuscript and addressed the majority of my concerns.

Reviewer #2 (Remarks to the Author):

The authors have addressed all my concerns with considerable care and effort, and have greatly improved their manuscript. I have no further concerns or comments and look forward to see the manuscript being published at Nature Communications.

Reviewer #3 (Remarks to the Author):

The authors have provided some useful additional information in the revised manuscript but I remain concerned that there are serious issues with some key claims of the paper.

i) A key experimental claim is that representation of sensory stimuli in ACC is suppressed when the stimuli are task irrelevant, with the primary evidence for this coming from analyses showing more accurate decoding of the stimuli when relevant compared to irrelevant. I expressed concern in my original review that this might reflect decoding of the animal's decision, rather than the stimuli itself, as the decision would be highly correlated with the stimulus when relevant but not when irrelevant. The authors have done some additional analyses to address this concern but in my view none of them are convincing:

Figure S3A-D does a decoding analysis considering only trials when the subject licked (so the decision variable is always the same), but does not compare accuracy when the stimulus is relevant vs irrelevant, so does not speak directly to the question of whether decoding accuracy is higher when the stimulus is relevant after controlling for the animal's decision.

Figure S3E-H is an analysis in which the difference in activity between go and no-go stimuli across the trial is projected onto the decision vector for a decoder trained to discriminate between these stimuli at a timepoint 0.25-0.75 seconds after stimulus onset. The idea is that activity in this timepoint is a pure stimulus-driven response 'where neither suppression, nor choice-related activity was present', such that any difference in this projection between relevant and irrelevant conditions later in the trial cannot be due to the decision, but rather due to suppression of the stimulus representation itself. However, it is entirely possible that decision related activity is already present in this 0.25-0.75 post-stimulus-onset time window, and indeed the projection is already substantially larger at this timepoint in the relevant than irrelevant condition in all subjects and both stimulus modalities -consistent with activity at this timepoint already reflecting the subject's decision and inconsistent with it being purely stimulus driven.

Figure S3I-L uses an encoding analysis (linear regression) to argue that the stimulus explains variance in the neural activity in addition to that explained by the decision. This is a step in the right direction but does not address the key question of whether the variance that is uniquely explained by the stimulus (after controlling for variance explained by the decision) is larger when the stimulus is relevant than when irrelevant.

Regarding the experimental claims in the paper, the additional information provided about the behaviour and the stats raise some new issues:

ii) I had not realised in the original review that in the recording sessions there is only a single block of each context (visual vs auditory relevant), such that one context occurs at the start of the

session and one at the end. This introduces a major confound in the claim that context is encoded by the ACC neurons (figure 3A) because if the ACC neurons represent any variable that changes systematically from start to end of session (e.g. the animals changing motivational state due to satiety) then this will make it possible to decode session start vs end, and hence make 'context' decodable. The same issue would arise if the spike sorting process causes any neurons to drop out or appear part way through the session due to drift in their waveforms, which is eminently possible. The best protection against this is through experiment designs in which variables of interest changes multiple times across the session to decorrelate them from possible slow changes in activity or drift. Given the limitations of the existing data the only way to make a convincing argument that the neurons genuinely encode context would be to show that the decoded context changes abruptly following the block transition rather than smoothly across the session. A convincing demonstration of this would further need to ensure that the data used to train the decoder came only from trials well away from the block transition.

iii) The additional information provided about the stats indicates that for many analyses, timepoints were used as the experimental unit, e.g. in figure 2K the N is respectively 600 and 1200 for the ACC and V1 data, comprising 150 timepoints for each subject. This is not acceptable statistical practice as timepoints are far from statistically independent, due to the strong autocorrelations in neural data, not to mention any smoothing applied during analysis.

The arguments in the theory section 'Context-gated attention in activity subspaces' still do not make sense to me. In the original review I raised the concern that the matrix M mapping the context input onto the stimulus activity subspace itself depended on the stimulus, which given this matrix was being interpreted as synaptic weights did not make sense. This remains an issue in the revised manuscript. To see this consider the equation on line 947 of the revised manuscript, which starts $d(c) = F(s + Mc)$ where d is the decision, s is the stimulus, c is the context, and M the matrix mapping the context onto the stimulus subspace. In this equation both the vector s and the matrix m are defined in terms of scalar elements a and v. However, the stimulus s clearly has to change from trial-to-trial, as different stimuli can be presented, whereas M is a matrix of fixed synaptic weights (see e.g. line 938) which do not change from trial-to-trial, raising a clear contradiction as the common elements a and v cannot both change from trial-to-trial and represent fixed synaptic weights. The clarity of this section would benefit from notation which clearly differentiated those variables that take different values from trial-to-trial from those which are fixed.

Response to reviewers

We thank the reviewer for their constructive follow-up comments. We have addressed these comments in an updated manuscript. Responses to the comments are provided in this 'response to reviewer' document, with clear indications to the updates in the revised manuscript.

Reviewer #3 (Remarks to the Author):

The authors have provided some useful additional information in the revised manuscript but I remain concerned that there are serious issues with some key claims of the paper.

i) A key experimental claim is that representation of sensory stimuli in ACC is suppressed when the stimuli are task irrelevant, with the primary evidence for this coming from analyses showing more accurate decoding of the stimuli when relevant compared to irrelevant. I expressed concern in my original review that this might reflect decoding of the animal's decision, rather than the stimuli itself, as the decision would be highly correlated with the stimulus when relevant but not when irrelevant. The authors have done some additional analyses to address this concern but in my view none of them are convincing:

Figure S3A-D does a decoding analysis considering only trials when the subject licked (so the decision variable is always the same), but does not compare accuracy when the stimulus is relevant vs irrelevant, so does not speak directly to the question of whether decoding accuracy is higher when the stimulus is relevant after controlling for the animals decision.

We performed the above mentioned analysis to test the hypothesis that responses late in the trial merely reflect decisions/movements instead of stimulus-related activity. Note that this analysis precisely addresses the reviewer's concern, which we explain here in detail. An alternative explanation was proposed, which stated that the apparent suppression in the irrelevant condition was not a consequence of suppression of the irrelevant stimulus modality, instead reflected the correlation of the stimulus identity with decision/movement. This is a highly relevant alternative interpretation since when the stimulus modality is relevant, the stimulus identity predicts the decision, thus a decision related activity would turn up in the decoding analysis as a component that identifies stimulus identity. To address this potential confound, we performed an analysis in which the decision/movement was the same when the two stimuli were presented. We argued that if the stimulus decodability resulted merely from decision/movement-related activity, then fixing the decision/movement would reduce stimulus decodability. The fact that a stimulus decoder is not distinguishable when the analysis is performed on all trials and when the analysis is constrained to trials in which lick always occurs highlights that

this interpretation does not hold, therefore different movement patterns are unlikely to account for the lower discriminability of stimuli in the irrelevant condition.

The Reviewer seems to propose an analysis in which the behavior is controlled AND the relevance of stimulus is changing. Unfortunately, these sets of conditions can only be fulfilled in exploratory trials, as the movement controlled analysis requires a sufficiently large number of error trials for reliable cross-validated decoding. These conditions can be fulfilled in two animals. Our earlier results indicated that exploratory trials are characterized by lower levels of suppression. Accordingly, we expected less pronounced suppression in this decision-controlled analysis. Our analysis has demonstrated that the irrelevant condition remained suppressed, in 3 out of 4 available context blocks in the two animals of the decision-controlled analysis. We updated Fig. S4A-D (formerly Fig. S3A-D), and applied corrected block averaged statistics (see below). These control analyses provide strong support for movement/decision related activity not being sufficient to account for suppression in the irrelevant condition. We expand on the original argument in the main text of the paper (pg. 8, lines 297-303)

Figure S3E-H is an analysis in which the difference in activity between go and no-go stimuli across the trial is projected onto the decision vector for a decoder trained to discriminate between these stimuli at a timepoint 0.25-0.75 seconds after stimulus onset. The idea is that activity in this timepoint is a pure stimulus-driven response 'where neither suppression, nor choice-related activity was present', such that any difference in this projection between relevant and irrelevant conditions later in the trial cannot be due to the decision, but rather due to suppression of the stimulus representation itself. However, it is entirely possible that decision related activity is already present in this 0.25-0.75 post-stimulus-onset time window, and indeed the projection is already substantially larger at this timepoint in the relevant than irrelevant condition in all subjects and both stimulus modalities -consistent with activity at this timepoint already reflecting the subject's decision and inconsistent with it being purely stimulus driven.

We agree with the reviewer that decision-related activity can appear early in the trial. However, we believe that this analysis provides useful insights as it provides evidence that neuronal response related movement (i.e. licking centered around the water availability time point at 2 sec with very rarely occurring early licking before 1 s into stimulus) not having a confounding effect on suppression. The original text element referred to this control as "We hypothesized that if the source of activity during suppression was different, e.g. movement induced activity...". We have updated this section in the main text for clarity (pg. 9, line 331).

Figure S3I-L uses an encoding analysis (linear regression) to argue that the stimulus explains variance in the neural activity in addition to that explained by the decision. This is a step in the right direction but does not address the key question of whether the variance that is uniquely explained by the stimulus (after controlling for variance explained by the decision) is larger when the stimulus is relevant than when irrelevant.

In the previous round of reviews we had demonstrated that stimulus-related variance exists in ACC neurons beyond the activity related to choice (current Fig. S4I-L, formerly S3I-L). As above, this analysis itself is insightful as it refutes the hypothesis that in the period when suppression is identified only choice-related activity is present. We have now updated this analysis to further address the t the concerns of the reviewer. We added an analysis that shows R^2 for the irrelevant stimulus modality. When neural activity was predicted from the irrelevant stimulus only, after suppression it had smaller R^2 than for relevant stimuli. This replicates the analysis in Fig. 2 by decoders. We added this single predictor to show general consistency between methods. The important addition answering the reviewer's question, though, is, when comparing R^2 from predictors of relevant stimulus + choice vs. irrelevant stimulus + choice. We only used trials from the consistent periods, as we evidenced on Fig. 2, that mice did not suppress irrelevant stimulus-related activity in exploratory trials. We changed the mean of R^2 over predictable neurons to best predictable neuron, as we found that it is better suited for demonstrating the existence of task related activity whilst also numerically correct for estimating R^2 . Since predicting a continuous variable from a binary task variable is generally much more difficult than using population decoders, the activity of many neurons was not predictable with cross-validation. In addition, the number of trials was small in multiple cases, similar to Fig. S4A-D, thus certain mice did not have even a single neuron whose activity was predictable from task variables in the consistent periods throughout the entire time course of trials. Unpredictable mice and contexts were left as blank. Those mice and contexts, however, where neuronal activity was predictable from task variables, showed the expected result: co-predictors of relevant stimulus + choice had significantly larger explained variance than irrelevant stimulus + choice. We applied the statistics correction based on the autocorrelation lag here as well, using 4 block-averaged time points for each mouse and context. Changes are reflected in Supplementary Fig. S4I-L and in the text (pg. 9-10, lines 348-362).

Regarding the experimental claims in the paper, the additional information provided about the behaviour and the stats raise some new issues:

ii) I had not realised in the original review that in the recording sessions there is only a single block of each context (visual vs auditory relevant), such that one context occurs at the start of the session and one at the end. This introduces an major confound in the claim that context is encoded by the ACC neurons (figure 3A) because if the ACC neurons represent any variable that changes systematically from start to end of session (e.g. the animals changing motivational state due to satiety) then this will make it possible to decode session start vs end, and hence make 'context' decodable. The same issue would arise if the spike sorting process causes any neurons to drop out or appear part way through the session due to drift in their waveforms, which is eminently possible. The best protection against this is through experiment designs in which variables of interest changes multiple times across the session to decorrelate them from possible slow changes in activity or drift. Given the limitations of the existing data the only way to make a convincing argument that the neurons genuinely encode context would be to show that the decoded context changes abruptly following the block transition rather than smoothly

across the session. A convincing demonstration of this would further need to ensure that the data used to train the decoder came only from trials well away from the block transition.

We thank the reviewer for highlighting their concerns. We have now added two extensions to the manuscript to address these issues.

First, although the Methods contained a brief description of drift control, we have now added a new paragraph (pg. 31, lines 1221-1230) that describes in more detail the systematic manual curation we originally used during spike sorting. These were designed rigorously and tested extensively to eliminate contributions from dropping out and appearing neurons.

Second, we have now implemented the suggested cross-validation scheme for context. It is fair to assume that it is difficult to distinguish the effect of transition between contexts in terms of model predictions: Both training and cross-testing in the transition trials can cause either worse or better predictions in the end. Therefore we assumed that comparing both training for context at the edges of the session and testing in the middle, and vice versa, training in the middle of the session and testing at the edges should yield similar context-accuracy time-courses, if drift is not present. We found that in all mice, this was indeed the case. This also corroborates the finding that differences in activity patterns between the two contexts are indeed equivalent within a context block and abruptly changes between the blocks. This control analysis is now added as a new Supplementary Fig. 3 on panels A-D for each mouse and corresponding text was added to the main text of the paper (pg. 10, lines 377-384).

iii) The additional information provided about the stats indicates that for many analyses, timepoints were used as the experimental unit, e.g. in figure 2K the N is respectively 600 and 1200 for the ACC and V1 data, comprising 150 timepoints for each subject. This is not acceptable statistical practice as timepoints are far from statistically independent, due to the strong autocorrelations in neural data, not to mention any smoothing applied during analysis.

We thank the reviewer for adding this point. To address this issue, we estimated the autocorrelation function of predictive decoder accuracy time courses. We found that the autocorrelation safely reached 0 in about 600 ms, with largely indistinguishable function profiles between conditions and animals. We used this window length to establish 4 time points during stimulus suppression between 0.6-3s, and used these as individual observation points for each mouse. The statistics of comparison between consistent and exploratory conditions in each brain area holds. We added the autocorrelation functions to Fig. 2 as panel K, and renamed the distribution comparison panel to L.

The arguments in the theory section 'Context-gated attention in activity subspaces' still do not make sense to me. In the original review I raised the concern that the matrix M mapping the context input onto the stimulus activity subspace itself depended on the stimulus, which given this matrix was being interpreted as synaptic weights did not make sense. This remains an issue

in the revised manuscript. To see this consider the equation on line 947 of the revised manuscript, which starts $d(c) = F(s + Mc)$ where d is the decision, s is the stimulus, c is the context, and M the matrix mapping the context onto the stimulus subspace. In this equation both the vector s and the matrix m are defined in terms of scalar elements a and v . However, the stimulus s clearly has to change from trial-to-trial, as different stimuli can be presented, whereas M is a matrix of fixed synaptic weights (see e.g. line 938) which do not change from trial-to-trial, raising a clear contradiction as the common elements a and v cannot both change from trial-to-trial and represent fixed synaptic weights. The clarity of this section would benefit from notation which clearly differentiated those variables that take different values from trial-to-trial from those which are fixed.

We thank the reviewer for taking the effort to double check on the consistency of notations. We agree that some confusing elements of the notation were still present. The main message is that v and α , elements of the input matrix, M , are defined as learned optimal suppression/enhancement levels that are fixed over the course of trials, and does not directly depend on the stimuli in the current trial. For further clarification to resolve the intelligibility issue of this section, we now amended the manuscript with a change of notation everywhere, where we missed it. We have now removed the explicit values of $-a$, v and $-v$, a , where they still remained. In the main text we now also refer to m being an element of a given stimulus subspace, with opposite direction as to the activity in that subspace for inhibition and same direction for enhancement (pg. 16, lines 597-599). This should clear up any confusion caused by notation. The actual elements of the mapping are later clarified as previously: learned mean suppression/enhancement levels. We made similar changes in the Appendix with concrete values (pg. 25, lines 975-976, 986-989). We changed the elements of M from $-a$ and $-v$ to $-v$ and $-\alpha$ everywhere, clearly signifying that they are not the same per trial stimulus related activity vectors, but average suppression levels suitable to be fixed connection weights in the matrix.

We hope that these clarification and notation improvements are now satisfactory for intelligibility, and once again thank the reviewer for the diligent observations regarding notation.

REVIEWERS' COMMENTS

Reviewer #1 (Remarks to the Author):

The authors revised the previous version of the manuscript to directly address the concerns raised. They address the concerns adequately and I think the manuscript has been strengthened. However the authors cannot entirely rule out some of the confounding factors highlighted by the reviewer. I suggest that the authors directly address some of the remaining and unresolved (and unsolvable given the current experimental design) weaknesses (eg i. lack of a substantial control in which the behavior/movement is controlled AND the relevance of stimulus is changing; ii. lack of an experimental control for a state change during the session) in the discussion.

Reviewer #2 (Remarks to the Author):

The authors have addressed all remaining concerns in a very thoughtful, thorough and convincing manner.

Response to reviewers

We thank the reviewer for their supporting comments.

Reviewer #1 (Remarks to the Author):

The authors revised the previous version of the manuscript to directly address the concerns raised. They address the concerns adequately and I think the manuscript has been strengthened. However the authors cannot entirely rule out some of the confounding factors highlighted by the reviewer. I suggest that the authors directly address some of the remaining and unresolved (and unsolvable given the current experimental design) weaknesses (eg i. lack of a substantial control in which the behavior/movement is controlled AND the relevance of stimulus is changing; ii. lack of an experimental control for a state change during the session) in the discussion.

We have now added two points to the discussion that address these issues. The added text is at the 8th and 9th paragraph of the discussion.